# Structural insight into small molecule action on Frizzleds

Paweł Kozielewicz [1], Ainoleena Turku [1], Carl-Fredrik Bowin [1], Julian Petersen [1], Jana Valnohova[1], Maria Consuelo Alonso Cañizal [2,3], Yuki Ono[4], Asuka Inoue[4], Carsten Hoffmann[2,3] & Gunnar Schulte [1]*

WNT-Frizzled (FZD) signaling plays a critical role in embryonic development, stem cell regulation and tissue homeostasis. FZDs are linked to severe human pathology and are seen as a promising target for therapy. Despite intense efforts, no small molecule drugs with distinct efficacy have emerged. Here, we identify the Smoothened agonist SAG1.3 as a partial agonist of $FZD_6$ with limited subtype selectivity. Employing extensive in silico analysis, resonance energy transfer- and luciferase-based assays we describe the mode of action of SAG1.3. We define the ability of SAG1.3 to bind to $FZD_6$ and to induce conformational changes in the receptor, recruitment and activation of G proteins and dynamics in FZD–Dishevelled interaction. Our results provide the proof-of-principle that FZDs are targetable by small molecules acting on their seven transmembrane spanning core. Thus, we provide a starting point for a structure-guided and mechanism-based drug discovery process to exploit the potential of FZDs as therapeutic targets.

[1] Section of Receptor Biology & Signaling, Department of Physiology & Pharmacology, Karolinska Institutet, S-17165 Stockholm, Sweden. [2] Institute of Pharmacology and Toxicology, University of Würzburg, Versbacher Str. 9, 97078 Würzburg, Germany. [3] Institute for Molecular Cell Biology, CMB-Center for Molecular Biomedicine, University Hospital Jena, Friedrich-Schiller University Jena, Hans-Knöll-Strasse 2, 07745 Jena, Germany. [4] Department of Pharmacological Sciences, Tohoku University, Sendai 980-8578, Japan. *email: gunnar.schulte@ki.se

G protein-coupled receptors (GPCRs) are membrane proteins, which constitute as much as 30% of all drug targets[1,2]. However, of the ~800 GPCRs in human only a small fraction is targeted by FDA-approved drugs leaving a large untapped, therapeutic potential in the remaining receptors[1]. The Class F (Frizzled; FZD) of GPCRs, which consists of ten FZD paralogues (FZD$_{1-10}$) and Smoothened (SMO) is critically involved in embryonic development, organogenesis, stem cell regulation, and in the development of diverse pathologies, such as different forms of tumors, fibrosis, bone disease, cardiovascular conditions, and neurological disease[3]. While there are several small molecules available that target SMO as agonists (SAG1.3, SAG1.5, and purmorphamine), inverse agonists (cyclopamine-KAAD), and neutral antagonists (vismodegib and SANT-1), no small molecules with clear-cut pharmacology have been identified targeting any FZD. Given their involvement in pathology, FZDs harbor a huge therapeutic potential and therefore, drugging FZDs has attracted substantial attention[4–6]. Interestingly, the crystal structure of FZD$_4$, which presents a ligand-free receptor inferred that development of small molecules targeting the core of FZDs can be virtually impossible given the hydrophilic nature of the binding pocket[7], a notion that has previously been challenged[8]. In addition, the concept of allosteric modulators has been explored with the small molecules FzM1 and FzM1.8, which were characterized as negative and ago-positive allosteric modulators, respectively, acting on the third intracellular loop (ICL3) of FZD$_4$ with low degree of selectivity[9,10].

The WNT family of lipoglycoproteins constitutes endogenous agonists for FZDs, activating the receptor through interactions with its extracellular cysteine-rich domain (CRD)[11]. Intracellularly, FZDs interact with Dishevelled (DVL), which is a signaling hub to mediate β-catenin-dependent and planar cell polarity (PCP)-like WNT signaling[12]. Furthermore, heterotrimeric G proteins interact with FZDs to initiate a network of G protein-dependent signaling pathways[5,13]. One of the explanations for the absence of FZD-targeting small molecule compounds is the lack of high-throughput assays that monitor FZD activation more directly than the T-cell factor/lymphoid enhancer-binding factor (TCF/LEF) transcriptional reporter (TopFlash) assay can do. The TopFlash assay has the clear disadvantages that (i) not all FZDs, particularly not FZD$_3$ and FZD$_6$, mediate WNT/β-catenin signaling and (ii) it does not cover all signaling pathways that branch off from activated FZDs, such as PCP or G protein-dependent signaling[5,14]. Recently developed resonance energy transfer-based methods (bioluminescence resonance energy transfer, BRET and Förster resonance energy transfer, FRET) can be advantageous to obtain more direct insight into FZD activation manifested in receptor conformational changes, FZD–G protein interaction, G protein activation, and FZD–DVL interactions[15–17].

Based on the sequence homology between SMO and FZD$_6$, and the possibility that SMO ligands could act on closely related FZDs, we show here that the small molecule SMO agonist SAG1.3 targets the transmembrane core of FZD$_6$ as a partial agonist with limited subtype selectivity. SAG1.3 binds FZD$_6$ and evokes a conformational change reminiscent of that seen in other agonist-bound GPCRs. Moreover, SAG1.3 stimulates FZD$_6$-dependent DVL membrane recruitment arguing that SAG1.3 stabilizes distinct receptor conformations accommodating G protein or DVL, supporting pathway-dependent functional selectivity[15]. In summary, our data indicate that FZDs can be targeted by small molecules.

## Results

### The SMO ligand binding pocket is similar in FZD$_6$. Phylogenetic analysis of Class F receptors[3,7,15] and sequence alignment of

human SMO, FZD$_6$, and FZD$_4$ indicate substantial homology among these receptors. However, FZD$_6$ shows a higher degree of sequence similarity with SMO than it does with FZD$_4$ at regions corresponding to the small-molecule binding pocket within the 7TM core (as observed in SMO; Fig. 1a and Supplementary Fig. 1), suggesting also functional similarities. Here, we chose to compare FZD$_6$ to FZD$_4$ and SMO because crystal structures of FZD$_4$ and SMO allow comparison on the atomistic level. Despite the differences that SMO mediates hedgehog signaling and FZD$_6$ mediates WNT signaling, both are characterized by their ability to couple to and activate heterotrimeric G$_i$ proteins[15,18–21], and their inability to signal via the WNT/β-catenin pathway[22,23]. FZD$_6$ and SMO are both characterized by a long TM6 extending above the plasma membrane toward the CRD[15,24–26] and the longest C-terminal domains across Class F receptors (SMO: 250 aa; FZD$_6$: 211 aa). Thus, the similarities of FZD$_6$ and SMO compared to FZD$_4$ provided the basis of our efforts of reprofiling SMO agonists for FZD$_6$.

**In silico analysis of SAG1.3–FZD$_6$ interactions.** As the putative small-molecule binding pockets in the transmembrane domain of FZD$_6$ are unknown, we built 15 homology models of FZD$_6$ using the ΔCRD SMO–taladegib (PDB ID: 4JKV) complex as a template[27]. Of these models, we selected the one with the best DOPE score for further studies[28]. The selected FZD$_6$ model (inactive FZD$_6$) subsequently underwent molecular dynamics (MD) simulations for 200 ns (in two independent replicas) in the ligand-free state in order to relax the structure. Subsequently, SAG1.3 (Fig. 1b) was docked to the binding site in the transmembrane core of the receptor, defined by the location of the cocrystallized SAG1.5 in the SMO crystal structure (PDB ID: 4QIN[29]; Fig. 1c). To compare SAG1.3–FZD$_6$ interactions with those present in a SAG1.3–SMO complex, we used the SAG1.5–SMO crystal structure, in which we modified the agonist by substituting the fluorine atoms for hydrogen atoms. Subsequently, the MD simulations were run for additional 3 μs (1 μs in three independent replicas) and 600 ns (200 ns in three independent replicas) with SAG1.3–FZD$_6$ and SAG1.3–SMO complexes, respectively (Fig. 1d, e, Supplementary Figs. 2 and 3).

Provided the recent insight into SMO activation in a ternary complex of ligand, receptor, and heterotrimeric G$_i$ protein (PDB ID: 6OT0[21]), we built also a FZD$_6$ model based on the active SMO structure and ran MD simulations with SAG1.3 docked to the same binding site as described above (Fig. 1d, e, Supplementary Fig. 3; active-like FZD$_6$). The MD data (three independent replicas of 1 μs each) were then used for a retrospective analysis of the binding site and interactions of SAG1.3 in complex with active-like FZD$_6$. To avoid misleading interpretations, we consider only one of the 1 μs replica of the simulations of the SAG1.3-bound inactive FZD$_6$ in all analyses; the N-terminus and extracellular loops in the other two replicas started to undergo rapid and noisy fluctuations after 400 ns and 600 ns of the simulations, respectively (most probably due to the instability in the homology model). With the active-like FZD$_6$ model, such behavior did not occur, and all 3 μs of data are considered in all analyses (Fig. 1d, e, Supplementary Fig. 3).

The overall binding location of SAG1.3 remained robustly similar in both studied proteins throughout different simulations (Fig. 1, Supplementary Figs. 2–4), suggesting that FZD$_6$ has a binding site for SAG1.3 in the transmembrane core of the receptor between the TM5, TM6, TM7, and the extracellular loop 2 (ECL2), similar to SMO. When comparing this binding area to the structure of FZD$_4$ (PDB ID: 6BD4, after 3 × 200 ns MD simulations), the extracellular portion of TM6 in FZD$_4$ together with the ECL3 clash heavily with the suggested SAG1.3 binding

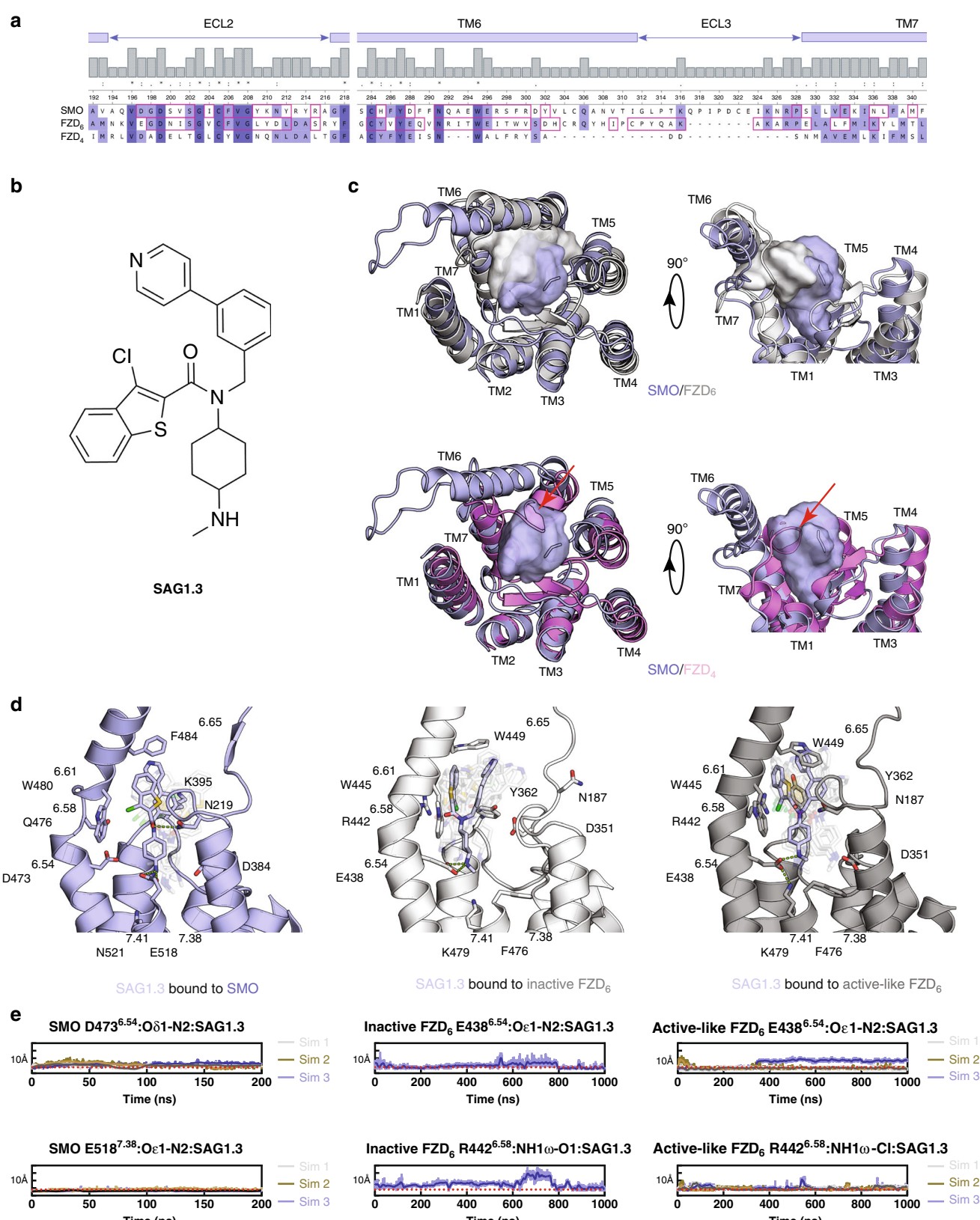

When zooming in to the details of ligand–receptor interaction, the SAG1.5-SMO crystal structure provides two main interactions, the D473[6.54]-amine nitrogen of SAG1.5 and the N219-amide oxygen of SAG1.5 (Ballesteros-Weinstein numbering of GPCRs[30]). MD simulations ($3 \times 200$ ns) suggest that binding of SAG1.3 to SMO is more versatile than binding of SAG1.5

site underlining a structural basis for ligand–receptor selectivity (Fig. 1c)[7]. Sequence alignment of these receptors supports this observation, since the parts of TM6 and ECL3 that construct the binding site of SAG1.3 in FZD₆ and SMO are for the most part missing from the sequence of FZD₄ (Fig. 1a, Supplementary Fig. 1).

**Fig. 1 The binding pocket of FZD$_6$ accommodates SAG1.3. a** Sequence alignments of the binding pockets of the human SMO, FZD$_6$, and FZD$_4$ (Supplementary Fig. 1 and Supplementary Data file 34). Red squares indicate residues in close proximity (<4 Å) between SAG1.3 and the receptor from the SMO and FZD$_6$ molecular dynamics (MD) simulations. **b** Structure of SAG1.3. The bold nitrogen represents the N2 referred to in the MD simulations below. **c** Comparison of the SAG1.3 binding sites of SMO and FZD$_4$ (inactive model; upper panel), and SMO and FZD$_4$ (lower panel) underlining the inability of FZD$_4$ to accommodate a ligand-like SAG1.3 in this binding space because of the short TM6 (red arrow). **d** The last frames from the selected MD simulations of the SAG1.3–SMO (left panel) and the SAG1.3–FZD$_6$ complexes (inactive model: middle panel, active-like model: right panel) with the important residues of the binding site depicted as sticks. Different positions of SAG1.3 throughout the time of simulation are indicated by transparent SAG1.3 molecules in the binding pocket. **e** Distance plots over simulation time between SMO D473$^{6.54}$–SAG1.3, SMO E518$^{7.38}$–SAG1.3, FZD$_6$ E438$^{6.54}$–SAG1.3, and R442$^{6.58}$–SAG1.3 (inactive and active-like models), which are predicted to form H-bonding interactions important for stabilizing the SAG1.3 binding conformation. The dotted line (red) indicates the maximum distance (4 Å) that is still likely to allow polar interactions. Thick traces indicate the moving average smoothed over a 2 ns window and thin traces represent raw data. The origin of the y-axis for all graphs **e** is 0 Å.

(Supplementary Fig. 2). The only structural differences between these two ligands are two fluorine atoms at the benzothiophene ring of SAG1.5 (Supplementary Fig. 2a), suggesting that this bulk restrains the conformational freedom of SAG1.5 in its binding site leading to a more stable binding pose to SMO (Supplementary Fig. 2b), which typically corresponds to tighter binding. This interpretation is in accordance with the previously published affinities of these two ligands. SAG1.5 showed ~2–10-fold higher affinity to SMO than SAG1.3[31,32]. In the SAG1.3–SMO MD simulations the complex of N219 with the amide oxygen remained in the hydrogen bonding distance (<4 Å) throughout the simulations (Supplementary Fig. 2c), whereas the protonated amine nitrogen (labeled as N2 throughout our study) of SAG1.3 was more often interacting with E518$^{7.38}$ than D473$^{6.54}$ (Fig. 1d, e, Supplementary Figs. 2d, e and 4). Additionally, D384 in ECL2 remains within 4 Å distance from N2 in approximately half of the MD frames (Supplementary Fig. 4).

In complex with the inactive FZD$_6$, N2 of SAG1.3 interacted quite robustly with E438$^{6.54}$ throughout the 1 μs simulation (Fig. 1d, e, Supplementary Figs. 3a and 4). The simulation with active-like FZD$_6$ strengthens this observation further. E438$^{6.54}$ remains at a hydrogen-bonding distance to N2 over 75% of all 3 μs of these MD trajectories (Fig. 1d, e, Supplementary Figs. 3d and 4). The rest of the time, N2 of SAG1.3 is interacting with D351 at ECL2 (D384 in SMO; Supplementary Fig. 3e and 4). Interestingly, the active-like FZD$_6$ simulation produced two clear clusters of binding poses, whereas in the inactive FZD$_6$ simulation the SAG1.3 poses—apart from the fact that they maintain N2–E438$^{6.54}$ interaction—were notably more deviant (Fig. 1d, Supplementary Fig. 3f, g). Unlike SMO, the FZD$_6$ binding site contains only these two negatively charged amino acid residues able to interact with N2 of SAG1.3; thus SAG1.3 does not change the charge-assisted hydrogen-bonding partner to amino acid 7.38 in FZD$_6$ as it does in SMO (Fig. 1a, d, Supplementary Fig. 4).

In the active-like FZD$_6$ simulations, the average distance between N187 of FZD$_6$ (corresponding to N219 of SMO) and the amide oxygen of SAG1.3 is ~5 Å (Supplementary Figs. 3h and 4). Even though the distance is most of the time too long to suggest direct hydrogen bonding, a water-mediated hydrogen bond is highly possible (Supplementary Fig. 3i). Unlike the active-like FZD$_6$ model, the inactive model rarely poses N187 in the vicinity of the amide oxygen of SAG1.3, but R442$^{6.58}$ remains there instead (Supplementary Fig. 3b, c). In the active-like model, R442$^{6.58}$ interacts rather with the chlorine atom of SAG1.3 (Fig. 1d, e).

In the SAG1.5–SMO crystal structure, the corresponding MD simulations, and the SAG1.3–SMO and SAG1.3–FZD$_6$ MD simulations, the two aromatic ends of the SAG derivatives form a stacked π–π complex located at a sub-pocket lined by aromatic amino acid residues of the TM6 and the ECL2 (Fig. 1a, d, Supplementary Fig. 4). Due to the different sizes of these residues (F484$^{6.65}$ in SMO vs. W449$^{6.65}$ in FZD$_6$), the aromatic end of SAG1.3 occupies a slightly different space and obtains a slightly

different orientation in these two receptors (Fig. 1d). As SAG1.3 is a relatively rigid molecule (only five rotatable bonds, which contribute to its overall shape), the orientation of the aromatic part of the molecule restricts the available locations of the hydrogen-bonding functional groups of SAG1.3 at its binding site. Even though the SAG1.3–SMO (based on the MD data), SAG1.5–SMO (based on the crystal structure and the MD data), and the SAG1.3–FZD$_6$ (based on the MD data) complexes share similar hydrogen-bonding characteristics (Supplementary Figs. 2–5), the apparent fit of the aromatic ends of SAG derivatives to these receptors may be one of the factors contributing to differences in affinity to SMO and FZD$_6$. Please see Supplementary Figs. 6 and 7, and Supplementary Data files 1–30 for the details of all the MD simulation runs.

**SAG1.3 binds to FZD$_6$.** Pharmacological analysis of ligand–receptor interactions is best studied using direct ligand binding experiments. Here, we employed a recently established assay format based on NanoBRET detection between a BODIPY-tagged ligand and a nanoluciferase (Nluc)-tagged receptor (Fig. 2a; summary of FZD$_6$ constructs in Supplementary Fig. 8)[33]. The interaction of BODIPY–cyclopamine with SMO allows thorough characterization of SMO-binding ligands[34]. Given the similarities of SMO and FZD$_6$ in the ligand binding pocket, we used BODIPY-cyclopamine as a probe for FZD$_6$. Moreover, in order to exclude endogenously expressed SMO as a confounding factor in the BODIPY-cyclopamine-based binding assay, we generated a ΔSMO HEK293 cell line devoid of this GPCR using CRISPR/Cas9 genome editing (Supplementary Fig. 9). BODIPY–cyclopamine binding to Nluc–FZD$_6$ resulted in monophasic and saturable concentration-dependent BRET signals (Fig. 2b; BODIPY–cyclopamine pK$_d$ ± s.d. = 6.3 ± 0.1; refer to Supplementary Fig. 10a for the assessment of cell membrane expression of the constructs and Supplementary Fig. 10b/Supplementary Data file 31 for FZD$_6$–BODIPY–cyclopamine docking poses). Additionally, BRET was dependent on donor expression levels and acceptor:donor ratio and was not directly proportional to the acceptor levels arguing for specificity of BODIPY–cyclopamine to Nluc–FZD$_6$ binding (Supplementary Fig. 10c). In competition experiments, increasing concentrations of unlabeled SAG1.3 decreased BODIPY–cyclopamine (300 nM) binding to Nluc–FZD$_6$ in a concentration-dependent manner (Fig. 2c, SAG1.3 pK$_i$ ± s.d. = 5.6 ± 0.1). Similarly, a fixed concentration of SAG1.3 (10 μM) right shifted the BODIPY-cyclopamine binding curve (Supplementary Fig. 10d, BODIPY–cyclopamine with SAG1.3 (10 μM) pK$_d$ ± s.d. = 5.8 ± 0.2) and the BODIPY–cyclopamine binding curve was right shifted in the presence of 10 μM unlabeled cyclopamine (Supplementary Fig. 10e). Importantly, the SAG1.3-induced reduction in BODIPY–cyclopamine/Nluc–FZD$_6$ NanoBRET was not due to a nonspecific effect on luminescence or fluorescence

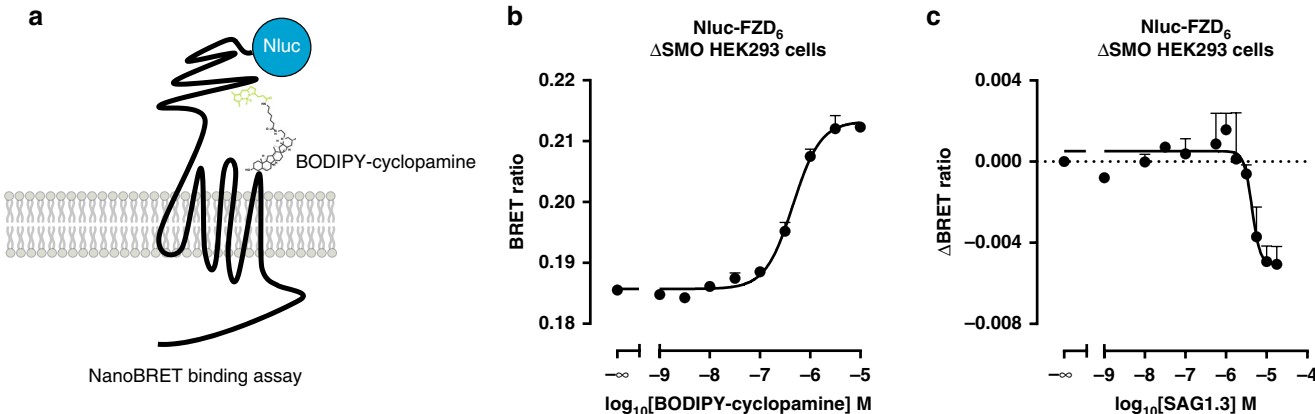

**Fig. 2 BODIPY–cyclopamine and SAG1.3 bind to FZD$_6$. a** The scheme depicts the experimental set up of NanoBRET analysis between the nanoluciferase-tagged FZD$_6$ and the BODIPY–cyclopamine. **b** BODIPY–cyclopamine induces a saturable concentration-dependent increase of BRET ratio. The graph presents raw NanoBRET values obtained following 90 min ligand exposure to living ΔSMO HEK293 cells. Data are presented as mean ± s.e.m. of total $n = 4$ individual experiments. **c** SAG1.3 displaces bound BODIPY–cyclopamine (300 nM) in a concentration-dependent manner. Data are presented as mean ± s.e.m. of total $n = 4$ individual experiments. Source data are provided as a Source Data file.

(Supplementary Fig. 10f). In addition, the NanoBRET-based binding assay has the advantage of a low contribution of nonspecific binding in general as seen for other GPCRs and in particular for Class F receptors, as we recently characterized during the establishment of the assay for Nluc–SMO and BODIPY–cyclopamine binding[34,35]. In conclusion, we provide here experimental data arguing that BODIPY–cyclopamine interacts with FZD$_6$ and that SAG1.3 interferes with BODIPY–cyclopamine interaction.

**A FZD$_6$–FRET probe monitors SAG1.3 binding and efficacy.** In order to obtain a functional measure for ligand–FZD$_6$ interactions, we designed an intramolecular FRET probe to study conformational changes of the receptor upon ligand binding. On the basis of conformational changes in active Class A and B GPCRs, the previously validated probes for other GPCRs and particularly FZD$_5$[16,36] and the information from the recently published active SMO structures[21,37] (Fig. 3a), we created an intramolecular FZD$_6$–FRET probe. The probe, which was designed to monitor agonist-induced conformational changes of FZD$_6$ in living cells, consists of the FRET donor (TFP) at the C-terminus and the FRET acceptor FlAsH (fluorescein arsenical hairpin binder-ethanedithiol, FlAsH–EDT$_2$)-binding motif inserted between G404 and R405 in the ICL3 (Fig. 3b).

The FZD$_6$–FRET sensor was detectable on the cell surface using confocal microscopy assessing TFP fluorescence, and we therefore conclude that it is efficiently trafficked to the cell membrane (Supplementary Fig. 11a). Basal energy transfer between the fluorophores was determined as FRET efficiency of the sensor (4.4 %) by using BAL (2,3-dimercapto-1-propanol) as an antidote (Supplementary Fig. 11b). In order to exclude energy transfer between individual receptors, for example in a FZD$_6$ dimer[25], we assessed intermolecular FRET between FZD$_6$–TFP and FZD$_6$–FlAsH–PK and detected no measurable energy transfer (Supplementary Fig. 11c).

To quantify the efficacy of the endogenous ligand of FZD$_6$ in this assay, we analyzed the effect of increasing concentrations of WNT-5A using the FZD$_6$–FRET probe. The maximal response to WNT-5A defines the full agonist in this assay with a $\log_{10}$EC$_{50}$ ± s.d. (ng ml$^{-1}$) = 2.2 ± 0.1 and the maximum efficacy at 1000 ng ml$^{-1}$ reaching 4.9% (FRET ratio = 95.1% of basal; Fig. 3c). Since WNT-5A is not membrane permeable, the effect of ligand stimulation on the FZD$_6$–FRET probe further

corroborates efficient trafficking of FZD$_6$–FlAsH–TFP to the plasma membrane. Stimulation of HEK293 cells transiently expressing the FZD$_6$–FRET probe with SAG1.3 resulted also in a sigmoidal, concentration-dependent decrease in the FRET ratio by 1.7 % of the basal FRET ratio with a pEC$_{50}$ ± s.d. (M) = 6.5 ± 0.9 (Fig. 3d).

Our data demonstrate that (i) SAG1.3 binds to FZD$_6$, (ii) that the polar residues D351, E438$^{6.54}$, and R442$^{6.58}$ have an important role in small-molecule binding, (iii) that agonist binding to FZD$_6$ evokes conformational changes that are detectable by the FZD$_6$–FRET sensor reminiscent of movements observed in activated Class A/B GPCRs and SMO, and (iv) that SAG1.3 acts as a FZD$_6$ partial agonist in this assay.

**SAG1.3 induces mini Gsi recruitment to FZD$_6$.** In order to further explore the mode of action of SAG1.3 on FZD$_6$, we made use of Venus-tagged mini G (mG) proteins, which serve as BRET-compatible, conformational sensors of the ligand-bound, active state of GPCRs[15,38,39] (Fig. 4a). Similar to what we have shown before for FZD$_6$ and other Class F receptors, we used SNAP–FZD$_6$–Rluc8 and FLAG–FZD$_6$–Nluc (BRET donor; see Supplementary Fig. 12 for analysis of membrane expression of FLAG–FZD$_6$–Nluc) in combination with Venus–mGsi (BRET acceptor) transiently overexpressed in HEK293 cells to monitor WNT-5A-induced Venus–mGsi recruitment to FZD$_6$, thereby defining the assay response with the physiological, full agonist (Fig. 4b, c). Further, we established the concentration–response relationship for both FZD$_6$ constructs in combination with Venus–mGsi using SAG1.3 (Fig. 4d, e). Interestingly, SAG1.3 induced a biphasic concentration–response curve similar to what was previously reported for SAG–SMO responses in the same assay format as well as in other assays[15,40–42]. In order to exclude a functional role of the WNT-binding CRD for the SAG1.3-induced and FZD$_6$-mediated Venus–mGsi recruitment, we compared ΔCRD and full-length FLAG–FZD$_6$–Nluc constructs (Fig. 4e; see Supplementary Fig. 12 for analysis of membrane expression of ΔCRD FLAG–FZD$_6$–Nluc). Irrespective of the presence or absence of the CRD, SAG1.3 evoked similar, concentration-dependent Venus–mGsi recruitment confirming the location of the SAG1.3 binding site in the receptor core and the irrelevance of the CRD for receptor-activating conformational changes.

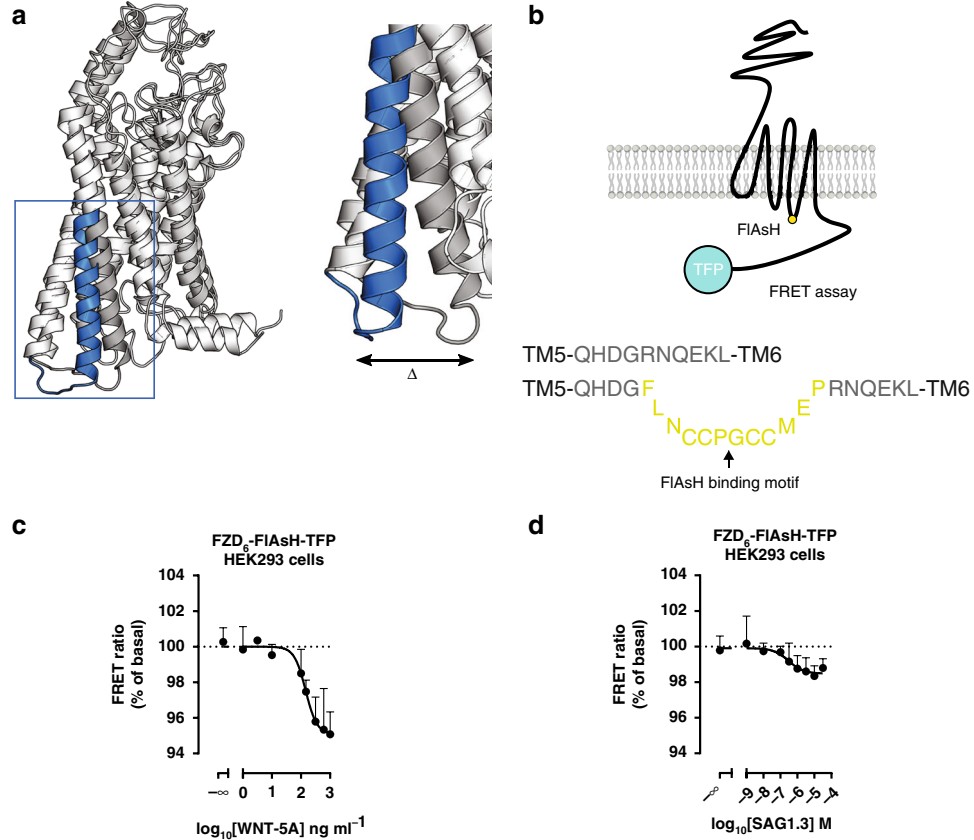

**Fig. 3 SAG1.3 induces conformational changes in FZD$_6$. a** The model of the active-like FZD$_6$ (blue) showing a pronounced outward-motion (Δ) of the TM6 as compared to the inactive model (gray), justifying positioning of FRET acceptor and donor in the ICL3 and C-terminus, respectively. **b** The scheme depicts the FZD$_6$–FlAsH–TFP construct. The FlAsH-binding motif (FLNCCPGCCMEP) was inserted into the ICL3, between G404 and R405. Receptor activation is predicted to result in a loss of FRET due to conformational rearrangement in accordance to previous data obtained for FZD$_5$[16]. **c** WNT-5A induced a concentration-dependent decrease of the FRET ratio (FlAsH/TFP) in HEK293 cells overexpressing FZD$_6$–FlAsH–TFP. The FRET ratio change induced by each concentration was normalized to basal FRET ratio. Data are represented as mean ± s.e.m. of total $n = 5$ individual experiments. **d** SAG1.3 induces a concentration-dependent decrease of the FRET ratio (FlAsH/TFP) in HEK293 cells overexpressing FZD$_6$–FlAsH–TFP. The FRET ratio change induced by each concentration was normalized to basal FRET ratio. Data are presented as mean ± s.e.m. of total $n = 7$ individual experiments. Source data are provided as a Source Data file.

**SAG1.3 action on FZD$_6$ is independent of SMO**. Since SAG1.3 was designed as SMO agonist, it appeared crucial to exclude a contribution of endogenously expressed SMO to the observed SAG1.3-induced and FZD$_6$-mediated effects. SMO is also a G$_{i/o}$-coupled receptor and it is expressed in HEK293 cells[15,20,43]. Thus, BRET mGsi recruitment assays were performed in the ΔSMO HEK293 cells using SNAP–FZD$_6$–$R$luc8 and Venus–mGsi in combination with increasing concentrations of SAG1.3. A similar biphasic concentration–response curve was observed (Fig. 4f). Furthermore, we compared ΔCRD and full-length FLAG–FZD$_6$–Nluc with regard to their ability to recruit Venus–mGsi in ΔSMO HEK293 cells (Fig. 4g), further supporting the concept that SAG1.3 targets the receptor core. In line with the results of the indirect binding assay with the FZD$_6$ intramolecular FRET sensor, SAG1.3 elicited a smaller maximum Venus–mGsi recruitment compared to the highest WNT-5A concentration used, underpinning the partial agonist nature of SAG1.3. In order to define subtype selectivity of SAG1.3 toward FZD$_6$ over FZD$_4$, we also assessed SAG1.3-induced Venus–mG recruitment to FZD$_4$–Nluc using Venus–mG13[15,44]. In agreement with the in silico structural analysis, which suggested that SAG1.3 would not bind this FZD subtype, we did not detect any SAG1.3-induced Venus–mG13 recruitment (Fig. 4h). On the other hand, we tested FZD$_7$, a Class F receptor from the FZD$_{1,2,7}$ homology cluster, and the ability of SAG1.3 to induce Venus–mGs

recruitment to SNAP–FZD$_7$–$R$luc8[15]. SAG1.3 induced a biphasic concentration–response curve similar to what we observed for FZD$_6$, indicating that SAG1.3 does not only act at FZD$_6$ but also on other FZD subtypes (Supplementary Fig. 13a). Indeed, the comparison of models of FZD$_6$ and FZD$_7$ on the atomistic level revealed large similarities in their SAG1.3 binding site (Supplementary Fig. 13b,c), in contrast to the one of FZD$_4$. MD simulation of FZD$_7$ bound to SAG1.3 further underlined that the receptor–ligand interaction is stable for the time of the simulation (Supplementary Fig. 13d).

**SAG1.3 promotes G protein and ERK1/2 activation**. In order to further validate that SAG1.3 acts as a functional FZD$_6$ agonist, capable of initiating downstream signaling in a G protein-dependent manner, we made use of heterotrimeric NanoBiT G proteins[45]. For this purpose, ΔSMO HEK293 cells were transfected with receptor or pcDNA, the Gα$_{i1}$ and Gβ$_5$ subunits fused to complementary parts of a modified Nluc (LgBiT and SmBiT) and the untagged Gγ$_2$ (Fig. 5a). First, we used the muscarinic M$_2$ receptor as a prototypical G$_i$-coupled receptor with acetylcholine (ACh), to demonstrate that we can detect a ligand-induced decrease in Nluc luminescence (Nluc$_{lum}$) indicative of the dissociation of the heterotrimeric G$_i$ protein (Supplementary Fig. 14, pEC$_{50}$ ± s.d. (M) = 7.3 ± 0.3; in agreement with http://

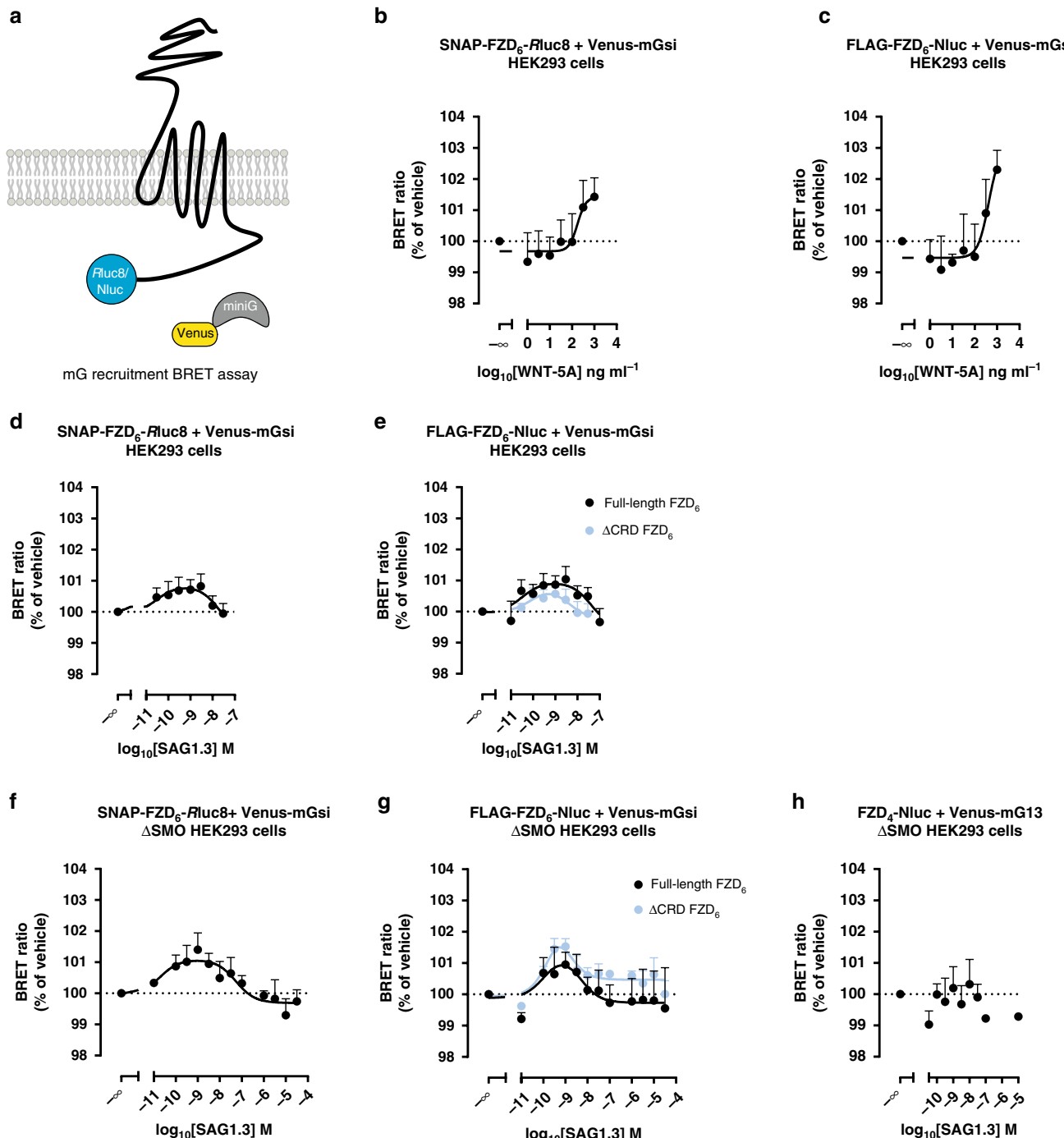

**Fig. 4 SAG1.3 mediates recruitment of mGsi proteins to FZD$_6$. a** The scheme depicts the experimental set up of BRET analysis between the luciferase-tagged FZD$_6$ and the Venus-tagged mGsi. Ligand stimulation initiates the mG protein recruitment to the receptor resulting in the increase of BRET. WNT-5A induced a concentration-dependent recruitment of the Venus–mGsi to SNAP–FZD$_6$–Rluc8 (**b**; total $n = 4$ individual experiments) and FLAG–FZD$_6$–Nluc (**c**; total $n = 3$ individual experiments) in transiently transfected HEK293 cells. SAG1.3 induced a bell-shaped, concentration-dependent recruitment of the Venus–mGsi to SNAP–FZD$_6$–Rluc8 (**d**; $n = 10$ individual experiments), FLAG–FZD$_6$–Nluc or ΔCRD FLAG–FZD$_6$–Nluc (**e**; total $n = 11$ individual experiments for FLAG–FZD$_6$–Nluc, and total $n = 8$ individual experiments for ΔCRD FLAG–FZD$_6$–Nluc) in transiently transfected HEK293 cells. **f** Similar experiments were performed in HEK293 lacking endogenous SMO (ΔSMO HEK293 cells). SAG1.3 showed concentration-dependent effects on SNAP–FZD$_6$–Rluc8 (**f**; total $n = 11$ individual experiments), FLAG–FZD$_6$–Nluc (**g**; total $n = 10$ individual experiments) and ΔCRD FLAG–FZD$_6$–Nluc-transfected (**g**; total $n = 8$ individual experiments) ΔSMO HEK293 cells. **h** SAG1.3 did not evoke Venus–mG13 recruitment to FZD$_4$–Nluc, which is consistent with the in silico prediction (total $n = 4$ individual experiments). All BRET data are presented as mean ± s.e.m. Source data are provided as a Source Data file.

www.guidetopharmacology.org/GRAC/ObjectDisplayForward?objectId = 14). Then we used SAG1.3 to monitor its ability to induce G$_i$ heterotrimer dissociation in a FZD$_6$-dependent manner, excluding the contribution of endogenous SMO by

using ΔSMO HEK293 cells. Similar to the Venus–mGsi protein recruitment assay, SAG1.3 elicited a bell-shaped concentration response only when SNAP–FZD$_6$ was coexpressed (Fig. 5b).

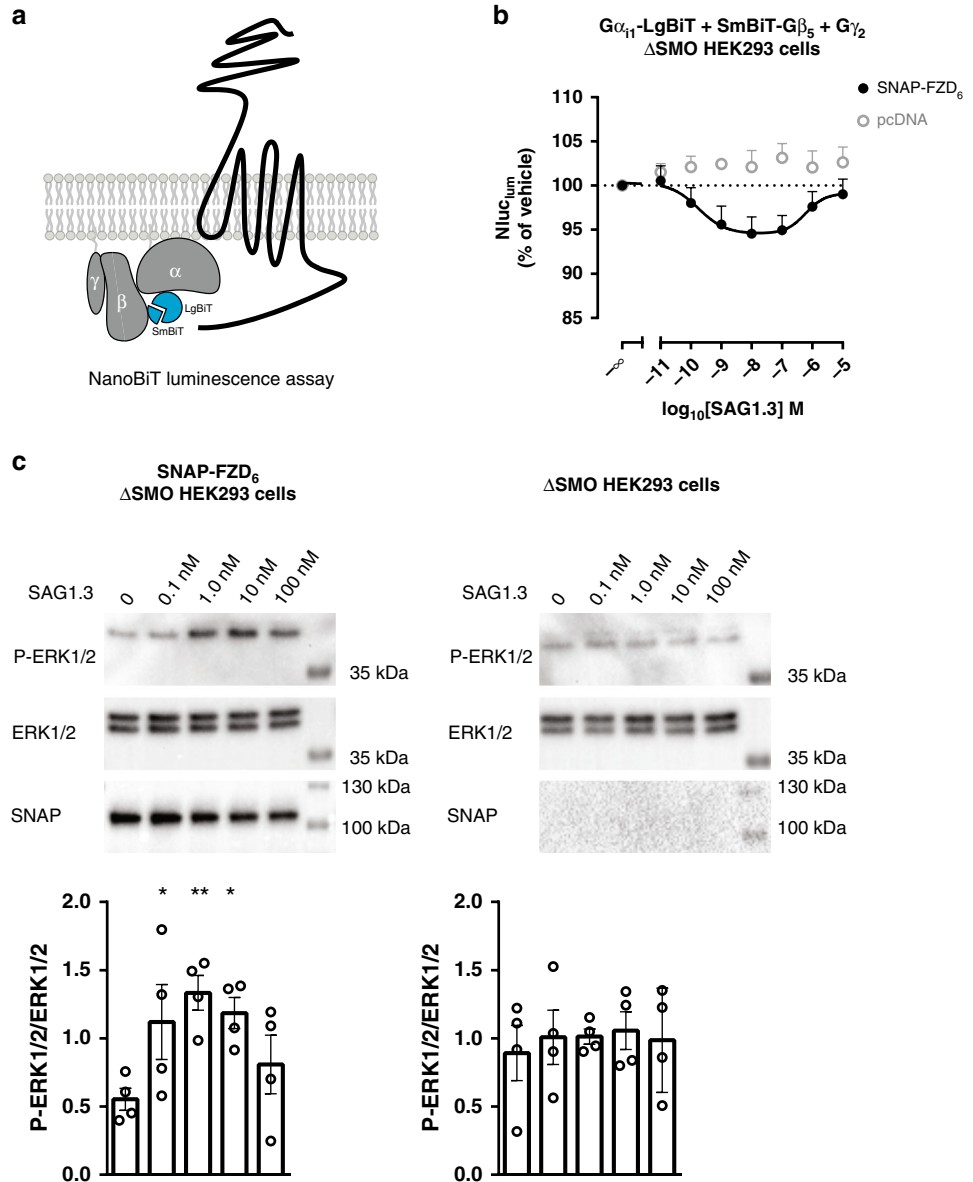

**Fig. 5 SAG1.3 induces FZD$_6$-dependent dissociation of heterotrimeric G$_i$ and phosphorylation of ERK1/2. a** Schematic view of the split NanoBiT luciferase assay. Ligand stimulation of a GPCR results in dissociation of the heterotrimeric G protein and a decrease in the Nluc luminescence. **b** SAG1.3 stimulation of SNAP–FZD$_6$ transiently overexpressed in ΔSMO HEK293 cells resulted in a concentration-dependent, biphasic decrease in basal luminescence as a measure of the dissociation of the Gα$_{i1}$-LgBiT, SmBiT–Gβ$_5$, and Gγ$_2$ complex (filled black circles). pcDNA served as no-receptor-control (open gray circles). Data are represented as mean ± s.e.m. of $n = 6$ individual experiments. **c** SAG1.3 (10 min) induced phosphorylation of ERK1/2 (P-ERK1/2) in a biphasic manner only in SNAP–FZD$_6$-transfected ΔSMO HEK293 cells. Serum starved cells were pretreated with C59 (5 nM; overnight). Representative immunoblots are shown. Data are presented as mean ± s.e.m. of $n = 4$ individual experiments; $F(4,14) = 3.141$. *$P < 0.05$, **$P < 0.01$ (one-way ANOVA). Source data are provided as a Source Data file.

GPCR-mediated activation of heterotrimeric G$_{i/o}$ proteins leads to phosphorylation and activation of extracellular signal-regulated kinases 1/2 (ERK1/2)[46] and we have previously shown that FZD$_6$ mediates ERK1/2 phosphorylation[15,25]. To further support the positive efficacy of SAG1.3 acting on FZD$_6$ resulting in G protein-dependent signaling, we quantified ERK1/2 phosphorylation in lysates of ΔSMO HEK293 cells transfected with SNAP–FZD$_6$ and stimulated with SAG1.3. These experiments were performed in the presence of endogenous G proteins. Further the autocrine stimulation by endogenously produced WNTs was blocked by pretreatment with the porcupine inhibitor C59 (5 nM). In agreement with our data so far, SAG1.3 induced a biphasic concentration-dependent ERK1/2 phosphorylation only

when FZD$_6$ was transiently overexpressed in ΔSMO HEK293 cells (Fig. 5c).

**SAG1.3 affects FZD$_6$–DVL2 interaction.** DVL is a central mediator of the β-catenin-dependent and PCP-like WNT signaling pathways and its recruitment to FZD is an initial step in DVL-dependent signaling[12,47,48]. Simultaneous overexpression of DVL and FZD leads to FZD-dependent membrane recruitment of DVL even in the absence of a ligand[49–51]. However, it remains obscure if and how WNT-mediated activation of FZDs affects this interaction dynamically. Investigation of the FZD–DVL interaction have previously relied on microscopic assessment of

colocalization and recruitment of cytosolic DVL present in punctate aggregates to membrane-expressed FZDs[47,49–51]. However, quantification of recruitment and measurement of ligand-induced dynamics were not possible. Employing direct BRET, it was recently shown that the FZD$_4$-selective agonist Norrin enhances FZD$_4$–DVL interaction[17]. In order to assess agonist-induced effects on FZD$_6$–DVL2 interactions, we used WNT-5A and SAG1.3 in two different experimental paradigms of BRET-based assays. First, we assessed the proximity of Nluc–DVL2 to SNAP–FZD$_6$ or FLAG–FZD$_6$–His indirectly in a bystander BRET assay[15]. Nluc–DVL2 membrane recruitment was quantified by co-expressing a membrane-bound Venus-tagged CAAX domain of KRas (termed Venus–KRas[52]), and assessment of bystander BRET between Nluc and Venus (Fig. 6a–d; Supplementary Fig. 15a)[15,39].

Second, we measured BRET between coexpressed Nluc–DVL2 and FLAG–FZD$_6$–Venus (Fig. 6e–g; Supplementary Fig. 15b; see Supplementary Fig. 12 for analysis of membrane expression of ΔCRD and full-length FLAG–FZD$_6$–Venus). The settings of the direct BRET assay are reverse to those employed recently[17], where the authors used YFP–DVL2 (BRET acceptor) and FZD$_4$–Rluc (BRET donor). However, it is envisaged that fusing the BRET acceptor to FZD circumvents the potential analysis issues arising from DVL polymerization at high expression levels required for validation of the assay by a saturation curve[53]. In order to assess ligand-induced effects on DVL–FZD BRET, we chose an acceptor:donor ratio corresponding to the plateau part of the saturation curves for both setups. In addition, we did not treat the cells with the porcupine inhibitor C59 to block secretion of endogenous WNTs as their presence had no significant effect on the basal recruitment of the overexpressed DVL2 to the overexpressed FZD$_6$ as recently reported[51] and presented in Supplementary Fig. 15a, b. To avoid any input of endogenous FZDs or SMO, we used ΔFZD$_{1–10}$ HEK293 cells to study WNT-5A-induced effects and ΔSMO HEK293 cells to study SAG1.3-induced effects. As shown in Fig. 6b, c, f, g, both ligands increased BRET between Nluc–DVL2 and Venus–KRas or FZD$_6$–Venus in a concentration-dependent manner. Interestingly, SAG1.3 evoked FZD$_6$–DVL2 BRET changes with lower potency than SAG1.3-induced G protein-related events. Moreover, SAG1.3 did not show the bell-shaped concentration–response when monitoring FZD–DVL recruitment. SAG1.3 also displayed a positive efficacy on ΔCRD FZD$_6$ (Fig. 6d, g). Further, we have validated the assays and SAG1.3-selectivity using pcDNA- and SNAP–FZD$_4$-transfected ΔSMO HEK293 cells (Fig. 6h). Biochemically, we were able to detect a SAG1.3-induced electrophoretic mobility shift of the endogenous DVL2 indicative of its phosphorylation and activation in the SNAP–FZD$_6$- but not control-transfected ΔSMO HEK293 cells (Supplementary Fig. 15c). Finally, we confirmed that SAG1.3 (10 μM) does not activate FZD$_4$-specific TopFlash activity in ΔFZD$_{1–10}$ HEK293 cells (Fig. 6i), arguing for the subtype selectivity of this small-molecule ligand.

WNT and SAG1.3 stimulation increased BRET in both experimental paradigms. However, given the ratio of receptor–DVL expression (Supplementary Fig. 14) and the nature of BRET as a readout, we cannot differentiate clearly between an increase in FZD–DVL recruitment in a 1:1 ratio, DVL polymerization in close proximity to the receptor or a rearrangement of the FZD–DVL complex in response to agonist affecting distance or dipole orientation. Nevertheless, a change in FZD–DVL BRET can serve as a functional readout of FZD ligands keeping the caveats of this technique in mind.

**Mutational analysis of the SAG1.3 binding site.** Having established a diverse set of functional readouts for FZD$_6$ activation by

SAG1.3 allowed now a mutagenesis analysis of residues involved in SAG1.3 interactions. The MD simulations in SAG1.5- and SAG1.3-bound SMO, and SAG1.3-bound FZD$_6$ using the inactive- and active-like models provided detailed insight into the engagement of residues in SAG–derivative interactions over time (Fig. 1d, Supplementary Fig. 4). In this analysis, D351, E438$^{6.54}$, K479$^{7.41}$, and R442$^{6.58}$ emerged as the most relevant, polar residues, which were included in a mutagenesis approach (Supplementary Fig. 16). SNAP-tagged receptor mutants were tested for their cellular and membranous expression in comparison to pcDNA- and SNAP–FZD$_6$-transfected ΔSMO HEK293 cells (Supplementary Fig. 16a). While these proteins are indeed translated, mutation of these residues dramatically affects receptor maturation and cell surface expression. Only E438D$^{6.54}$, R442A$^{6.58}$, and R442K$^{6.58}$ were detectable at the membrane albeit at lower levels compared to wild-type SNAP–FZD$_6$ (Supplementary Fig. 16a). Nevertheless, surface expression mirrored the ability of the receptor mutants to recruit Nluc–DVL2 to the membrane assessed by bystander BRET using Nluc–DVL2 and Venus–KRas (Supplementary Fig. 16b). In order to provide biologically sound and meaningful data, we only used three mutants showing surface expression and DVL recruitment to assess SAG1.3-induced effects. Mutation of the SAG1.3 binding site in FZD$_6$ affected the ability of SAG1.3 to induce mGsi and Nluc–DVL2 recruitment and G$_i$ protein activation (Supplementary Fig. 16c, d, e). In general, SAG1.3-induced responses of E438D$^{6.54}$ and R442K$^{6.58}$ mutants showed lower efficacy and potency when compared with the wild-type receptor (Supplementary Figs. 16c, d, e and 17). Furthermore, SAG1.3 stimulation of the R442A$^{6.58}$ mutant resulted in hardly detectable responses corroborating even weaker SAG1.3 interactions. Thus, mutating residues to their chemically conserved counterparts allows preserving a somewhat functional SAG1.3 binding site, whereas alanine mutation does not. While functional assays are directly affected by the fraction of the receptor protein that is trafficked to the cell membrane, we reasoned that assessment of ligand affinity could be a suitable complement to quantify the direct involvement of key residues in the FZD$_6$ binding site. We focused on FZD$_6$ R442$^{6.58}$ and its alanine mutation comparing the affinity of BODIPY–cyclopamine to FZD$_6$ and FZD$_6$ R442A$^{6.58}$ in the presence and absence of 10 μM SAG1.3 (Supplementary Fig. 18). Since the presence of SAG1.3 right shifted BODIPY–cyclopamine binding only in the case of wild-type FZD$_6$, we concluded that R442$^{6.58}$—in agreement with the in silico predictions and functional assessment—is a key component of the SAG1.3 binding site.

In order to further support our findings concerning the importance of N2–E438$^{6.54}$ interaction, we provide MD data investigating the likelihood of interaction for inactive FZD$_6$ with a mock SAG1.3 (mock ligand), where we introduced a carbon (C7) instead of the N2 nitrogen (Supplementary Fig. 19a). MD simulations of FZD$_6$ with SAG1.3 in comparison to mock ligand (3 × 200 ns) and distance plots between the E438$^{6.54}$ and either N2 of SAG1.3 or the C7 of the mock ligand indeed argue that SAG1.3 binds closer to E438$^{6.54}$ than the mock ligand over the time course of the simulation (Supplementary Fig. 19b).

**Purmorphamine is also a FZD$_6$ agonist.** In addition to SAG derivatives, purmorphamine presents another, structurally unrelated SMO agonist that is surmountable by the inverse agonist cyclopamine-KAAD (Supplementary Fig. 20a)[54,55]. In order to support the broader applicability of our findings, we also examined purmorphamine–FZD$_6$ interaction by in silico docking and performed a pharmacological characterization of purmorphamine activity using two key assays presented in this work. In

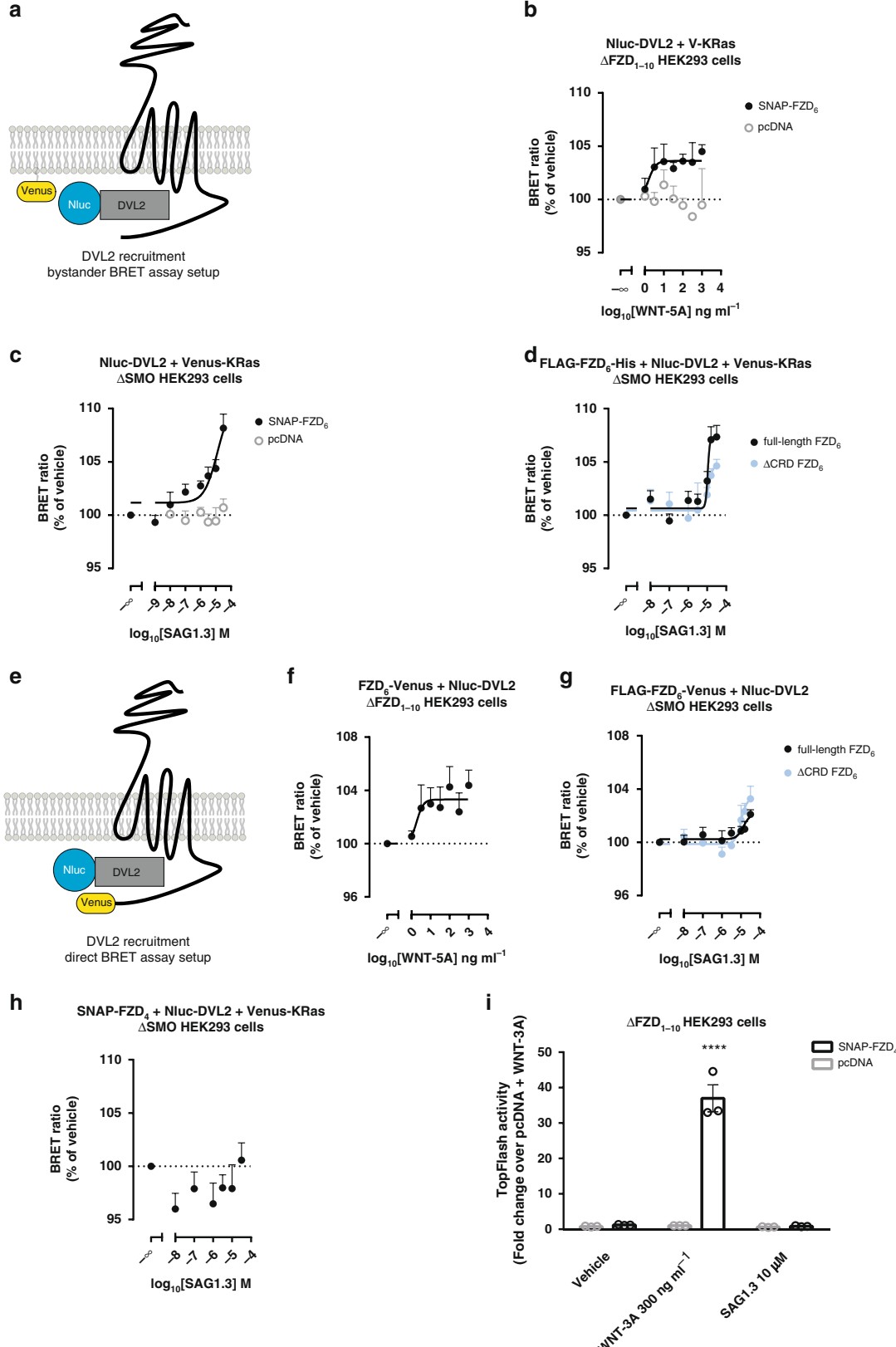

silico docking indicated that the purmorphamine binding site overlaps substantially with that of SAG1.3 (Supplementary Fig. 20b), in agreement with previous pharmacological characterization[55]. Compared to SAG1.3 purmorphamine promoted a similar, concentration-dependent recruitment of Venus–mGsi

to SMO–Rluc8 and SNAP–FZD$_6$–Rluc8, albeit with lower potency and without the distinct bell-shaped pattern (Supplementary Fig. 20c, d). Since the mGsi protein serves as sensor of the active FZD$_6$ conformation feeding into heterotrimeric G protein signaling, we conclude that purmorphamine binding to

**Fig. 6 SAG1.3 modifies the interactions between FZD$_6$ and DVL2. a** Schematic illustration of the bystander BRET setup to detect SNAP–FZD$_6$-induced recruitment of Nluc–DVL2 to membrane bound Venus–KRas. **b** WNT-5A stimulation of SNAP–FZD$_6$-transfected ΔFZD$_{1-10}$ HEK293 cells increased the bystander BRET ratio in a concentration-dependent manner (filled black circles; total $n = 5$ individual experiments). WNT-5A did not affect BRET in cells transfected with pcDNA (open gray circles; total $n =$ three individual experiments). **c** Bystander BRET ratio changes (Nluc–DVL2 and Venus–KRas) assessed in ΔSMO HEK293 cells in response to increasing concentrations of SAG1.3 in the presence of SNAP–FZD$_6$ (filled black circles; total $n = 9$ individual experiments) or pcDNA (open gray circles; $n = 5$ individual experiments). **d** Bystander BRET ratio changes (Nluc–DVL2 and Venus–KRas) assessed in ΔSMO HEK293 cells in response to increasing concentrations of SAG1.3 in the presence of FLAG–FZD$_6$–His (total $n = 6$ individual experiments) or ΔCRD FLAG–FZD$_6$–His (total $n = 6$ individual experiments) in the ΔSMO HEK293 cells. **e** The scheme illustrating the direct BRET setup in which the signal is detected between FLAG–FZD$_6$–Venus and Nluc–DVL2. **f** WNT-5A induced BRET ratio indicative of closer interactions between FLAG–FZD$_6$–Venus and Nluc–DVL2 (total $n = 6$ individual experiments) in the ΔFZD$_{1-10}$ HEK293 cells. **g** SAG1.3 induced BRET indicative of closer interactions between FLAG–FZD$_6$–Venus (total $n = 11$ individual experiments) or ΔCRD FLAG–FZD$_6$–Venus (total $n = 6$ individual experiments) and Nluc–DVL2 in the ΔSMO HEK293 cells. **h** SAG1.3 did not induce the bystander BRET ($n = 4$ individual experiments) or **i** TOPFlash reporter activity in the ΔSMO HEK293 cells with transiently overexpressed SNAP–FZD$_4$ (WNT-3A used as a positive control; $n = 3$ individual experiments; $F_{(2,12)} = 88.69$. ****$P < 0.0001$, two-way ANOVA). Data are presented as mean ± s.e.m. Source data are provided as a Source Data file.

---

FZD$_6$ results in receptor activation and G protein activation. On the other hand, purmorphamine did not affect FZD$_6$–DVL2 dynamics (Supplementary Fig. 20e) arguing for a distinct functional selectivity of this ligand.

## Discussion

Here, we provide the proof-of-principle that FZDs are druggable with small-molecule ligands targeting the 7TM core of the receptors. This stands in stark contrast to previous claims that the FZD binding pocket might be unfavorable for accommodating small-molecule ligands[7]. Our discovery opens the door for the development of FZD-targeting small molecules interacting with the receptor at a site reminiscent of that of Class A GPCRs and SMO ligands. Based on the data monitoring receptor binding, FZD$_6$ conformational changes, mG protein association as conformational sensors of the active GPCR state of FZD$_6$, heterotrimeric G protein dissociation, and FZD$_6$–DVL2 recruitment, we argue that SAG1.3 acts as a partial agonist with functional selectivity toward G proteins over DVL. SAG1.3-induced effects on FZD$_6$ were generally moderate but statistically significant calling for medicinal chemistry efforts to expand on our proof-of-concept study (Supplementary Fig. 21). We also provide evidence that SAG1.3 acts at FZD$_7$ but not FZD$_4$ and that purmorphamine acts through FZD$_6$, albeit with lower potency, indicating that different scaffolds exist to initiate a medicinal chemistry optimization. Furthermore, we use BODIPY–cyclopamine for assessing ligand binding introducing another, sterol-based moiety interacting with FZD$_6$. While small-molecule agonists will provide an exciting tool to understand FZD activation mechanisms and receptor pharmacology, ligands with negative or no efficacy would be more suitable for anticancer therapy. Inverse agonists or neutral antagonists could provide a useful therapeutic approach in tumors that are driven by high levels of WNTs or constitutively active FZDs[3,4,15].

In line with what we have previously proposed[5,15], our data suggest that SAG1.3 can stabilize at least two distinct FZD$_6$ conformations feeding into FZD–G protein and FZD–DVL signaling. This is supported by the finding that an active SAG1.3–FZD$_6$–mGsi complex is stabilized at lower nanomolar concentrations, whereas micromolar SAG1.3 concentrations are required to affect FZD$_6$–DVL2 interaction. Additional support of this concept is provided by the purmorphamine data, showing a positive efficacy toward mGsi protein but not DVL. Furthermore, SAG1.3 is not merely an allosteric modulator of FZD$_6$ amplifying basal WNT input as it significantly increases the phosphorylation of ERK1/2 in the absence of endogenous WNTs.

Employing in silico analysis, we predict that D351, E438$^{6.54}$, and R442$^{6.58}$ in FZD$_6$ are key residues involved in polar interactions with SAG1.3 (Fig. 1d). We then generated mutants of these

residues (together with K479$^{7.41}$) and assessed their cellular and membranous expression as well as SAG1.3-induced effects. SAG1.3-induced and receptor-mediated effects were reduced as shown by mGsi and Nluc–DVL2 BRET recruitment and G$_i$ NanoBiT assays, however, only few mutants are folded and trafficked to the cell membrane albeit not as efficiently as the wild-type FZD$_6$. This obviously needs to be taken into consideration when interpreting these mutagenesis experiments. BRET experiments are ratiometric, i.e., they do not rely on expression levels per se because the signal is detected only when donor and acceptor come into close proximity (up to 100 Å). Thus, BRET experiments can be used to evaluate and compare ligand-induced effects on non-equally expressed receptors. However, in the case of G$_i$-dissociation measured by the NanoBiT assay, it cannot be ruled out that—given the cell permeability of SAG1.3—events from inner membranes contributes to the response.

Furthermore, we have identified a polar network of interactions involving ECL2, TM6, and TM7 that is a part of the proposed FZD$_6$ ligand binding site. According to our mutagenesis data, this network is crucial to maintain proper protein folding and cell surface trafficking of FZD$_6$ (Supplementary Fig. 19c)[34,56,57].

The pharmacological profiles of SAG1.3 as an agonist on FZD$_6$ differ depending on the experimental readout. While SAG1.3-induced conformational changes in the FZD$_6$–FRET probe and FZD$_6$–DVL2 interaction assays follow a sigmoidal concentration–response relationship with an EC$_{50}$ in the lower micromolar range, all G protein-related readouts follow a bell-shaped pattern with higher potency. Remarkably, the shape of the concentration–response curves is similar to what has been reported for SAG1.3-induced and SMO-mediated effects on Gli-1 reporter, mG recruitment, and inositol phosphate accumulation assays[15,32,40,41]. However, it remains obscure what the descending part of these SAG1.3 concentration–response curves represents in the context of SMO and FZD$_6$ signaling[40,58], and interestingly, purmorphamine, which occupies a similar binding site, does not exert a bell-shaped concentration–response curve as SAG1.3 in the G protein-dependent assays. Similarly to what we demonstrate for SAG1.3–FZD$_6$, it can also be seen for SMO that SAG1.3 potencies differ depending on the assay type and the cell line used (Supplementary Fig. 22). While this could merely represent the differential functionality and sensitivity of the engineered assay probes, it seems more likely that they underline the functional selectivity or ligand bias of the agonist. Similar discrepancies in the pharmacological profiles have been reported for the β$_2$ adrenergic receptor[59].

The subtype selectivity of SAG1.3 toward FZD$_6$ and FZD$_7$ over FZD$_4$ is mostly determined by the length of TM6, which in case of a shorter connection to TM7 in FZD$_4$ traverses through the SAG1.3 binding pocket. While more experiments are necessary to

map the SAG1.3 selectivity for all FZD$_{1-10}$, it is likely that FZDs with a long TM6 could accommodate this ligand, whereas the homology cluster FZD$_{4,9,10}$ might not.

In summary, repurposing of a SMO agonist led to the identification of a partial agonist of FZD$_6$ that acts in the 7TM core of the receptor. Medicinal chemistry efforts are now required to define structure–activity relationships of FZD small-molecule ligands to better understand subtype selectivity, and to promote the discovery of diverse pharmacological probes for FZDs as a starting point for the development of therapeutic compounds.

## Methods

**Computational modelling and MD simulations.** The homology modeling of the inactive FZD$_6$ has been described previously[25]. Briefly, the taladegib-bound structure of SMO was used as a template (PDB ID: 4JKV)[29] and the sequence of FZD$_6$ (UniProt ID: O60353) was aligned to that of SMO (UniProt ID: Q99835) with ClustalX2[60]. The N- and C-termini were excluded due to a lack of suitable template, and the alignment was manually edited to ensure the proper alignment of transmembrane domains and conserved motifs present in Class F GPCRs. Fifteen homology models were generated with MODELLER 9.11[61] and a representative model was selected based on DOPE score and visual inspection.

Active—24,25(S)-epoxycholesterol bound and G$_i$ bound—SMO structure (PDB ID: 6OT0)[21], was published during the revision of our study. To implement these new structural data, we built 20 new FZD$_6$ and FZD$_7$ models with MODELLER 9.19[61] using the active SMO structure as a template. To select a representative model, SAG1.3 was docked to all these models with Glide software in the Schrödinger Release 2018-4 Maestro molecular modeling platform to a $20 \times 20 \times 20$ Å$^3$ box located based on the binding site of SAG1.5 in the SMO crystal structure (PDB ID: 4QIN). The model producing SAG1.3 docking poses best resembling the pose of SAG1.5 in complex with SMO was selected.

To obtain a starting pose of SAG1.3 in the inactive FZD$_6$ model, which would then be used in MD simulations, SAG1.3 was docked into the inactive FZD$_6$ model following 200 ns MD relaxation (see below) with AutoDock Vina[62] using a $20 \times 20 \times 20$ Å$^3$ box positioned on the location of SAG1.5 in the cocrystallized SAG1.5–SMO complex (PDB ID: 4QIN). To generate an initial SAG1.3–SMO complex, the fluorine atoms of SAG1.5 were substituted for hydrogens in the SAG1.5–SMO complex (PDB ID: 4QIN). The highest scoring pose of the SAG1.3–FZD$_6$ (inactive) docking experiments, and the SAG1.3–SMO structure were then used to initiate the MD simulations. The ligands were protonated at pH = 7.4 in Avogadro and their parameters were generated using CGenFF[63], and atomic charges evaluated using the Force Field Toolkit plugin in VMD[64].

The docking study of purmorphamine was conducted with AutoDock Vina similarly to SAG1.3, with both target proteins (inactive FZD$_6$ model or SMO crystal structure) relaxed by 200 ns of MD (see below) prior to docking. Docking study of BODIPY–cyclopamine was conducted with Glide, and LigPrep (Schrödinger Release 2018-4) with Epik was used for generating ligand conformations and protonation states (at pH 7 ± 2). The conformational complexity of BODIPY–cyclopamine was reduced by restraining the cyclopamine core to a similar conformation with the cyclopamine that is cocrystallized with SMO (PDB ID: 4O9R)[65]. The BODIPY–cyclopamine was docked to the same FZD$_6$ model as purmorphamine, to a $20 \times 20 \times 20$ Å$^3$ box located based on the SAG1.5 binding site in SMO as described above.

The best representatives of the active-like SAG1.3–FZD$_6$ and SAG1.3–FZD$_7$ docking complex was also used for initiating MD simulations. SAG1.3 was parametrized with AmberTools18 package (University of California San Francisco) using GAFF2 force field and AM1-BCC charges.

MD simulations were performed on the models of the inactive and active-like FZD$_6$, active-like FZD$_7$, the crystal structures of FZD$_4$, and SMO (PDB IDs: 6BD4[7] and 4QIN[29], respectively) using GROMACS[66]. The missing residues in the SMO structure (aa 434–440 and aa 494$^{6.75-505}$) were modelled using the SMO structures with PDB IDs: 4JKV and 5L7D[67], respectively. The missing residues in the FZD$_4$ structure (aa 420$^{5.76}$–427) were modelled using the SMO structure (PDB ID: 4JKV). The protonation states were assigned at pH = 7.4 in Chimera[68]. The OPM database was used to correctly orientate the proteins and the CHARMM-GUI server[69] was used to embed them in the phosphatidylcholine lipid bilayer, add water molecules and 0.15 M NaCl. Typically, the system was minimized in 1500 steps and was subsequently subjected to equilibration with gradually decreasing position restraints on protein and lipid components. In the last 50 ns of the equilibration run, the harmonic force constants of 50 kJ mol$^{-1}$ nm$^{-2}$ were applied on the protein and ligand atoms only. Lastly, the independent isobaric and isothermic (NPT) ensemble production simulations for each system were initiated from random velocities. In these simulations, the CHARMM36m force field was used with a 2 fs time step. The temperature at 310 K was maintained with Nose–Hoover thermostat and the pressure at 1 bar was maintained with Parrinello–Rahman barostat. Particle-mesh Ewald for electrostatic interactions and a 9 Å cutoff for van der Waals interactions were used. All the bonds between hydrogen and other atoms were constrained using the LINCS algorithm. The data

files were saved every 100 ps. The MD simulation data (~12.5 μs combined) were analyzed using VMD (analysis extensions "RMSD Trajectory Tool", "VolMap Tool", and "Hydrogen Bonds") and PyMol (The PyMOL Molecular Graphics System, Version 2.0 Schrödinger, LLC). The distance plots were produced with distance.tcl script (https://www.ks.uiuc.edu/Training/Tutorials/vmd/vmd-tutorial-files/distance.tcl). Please see the Supplementary Fig. 7 for more detailed information on the MD simulations. Snapshots of the MD simulations are provided as Supplementary Data files 1–34.

**Cell culture and ligands.** HEK293 cells (ATCC), ΔSMO HEK293A (generated in this study, please see below), and ΔFZD$_{1-10}$ HEK293T cells[70] were cultured in Dulbecco's modified Eagle's medium (DMEM) supplemented with 10% fetal bovine serum, 1% penicillin/streptomycin, and 1% L-glutamine (all from Thermo Fisher Scientific) in a humidified CO$_2$ incubator at 37 °C. All cell culture plastics were from Sarstedt, unless otherwise specified. Absence of mycoplasma contamination was routinely confirmed by PCR using 5′-GGCGAATGGGTGAGTAACACG-3′ and 5′-CGGATAACGCTTGCGACTATG-3′ primers detecting 16 S ribosomal RNA of mycoplasma in the media after 2–3 days of cell exposure. C59 (2-[4-(2-methylpyridin-4-yl)phenyl]-N-[4-(pyridin-3-yl)phenyl]acetamide; Abcam #ab142216; stored as 5 mM solution in aliquots in dimethyl sulfoxide (DMSO; at −20 °C) was used to inhibit porcupine to abrogate endogenous secretion of WNTs[71]. For stimulation, recombinant WNT-5A (R&D Systems/Biotechne #645-WN), recombinant WNT-3A (R&D Systems/Biotechne #5036-WN), SAG1.3 (3-chloro-N-[trans-4-(methylamino)cyclohexyl]-N-[[3-(4-pyridinyl)phenyl]methyl]-benzo[b]thiophene-2-carboxamide dihydrochloride; Sigma SML1314), purmorphamine (9-cyclohexyl-N-[4-(4-morpholinyl)phenyl]-2-(1-naphthalenyloxy)-9H-purin-6-amine; Abcam #ab120933), and acetylcholine (ACh; Sigma #A6625) were used. SAG1.3 was dissolved in water at 10 mM or DMSO at 100 mM and stored in aliquots at −20 °C. Purmorphamine was dissolved in DMSO at 10 mM and stored in aliquots at −20 °C. ACh was dissolved in water at 100 mM and stored in aliquots at −20 °C. Cyclopamine (Abcam #ab120392) was dissolved in DMSO at 1 mM and stored in aliquots at −20 °C. BODIPY–cyclopamine (BioVision #2160) was dissolved in DMSO at 1 mM and stored in aliquots at −20 °C. The ligands underwent a maximum of two freeze-thaw cycles. WNT-3A and WNT-5A were dissolved at 100 μg ml$^{-1}$ in filter-sterilized 0.1% bovine serum albumin/phosphate buffered saline (BSA/PBS) and stored at 4 °C. In the experiments with WNTs, plates and tips coated with Sigmacote (Sigma), and protein-low binding tubes (Eppendorf) were used to make serial dilutions and dispense the ligands. The experiments with BODIPY–cyclopamine were performed under low light conditions and the serial dilutions were made in the protein-low binding tubes.

**Generation of the ΔSMO HEK293A.** The ΔSMO HEK293A cells were generated by introducing random, frame-shift mutations in the SMO gene using a CRISPR/Cas9 system as described previously[72,73] with minor modifications. With the online CRISPR design tool (http://crispr.mit.edu), we selected the following SMO-targeting single guide RNA (sgRNA) construct that contained a restriction enzyme-recognizing site encompassing three-base pair upstream (SpCas9-mediated double-strand DNA cleavage position) of the SpCas9 PAM sequence (NGG): 5′-CAACCCCAAGAGCTG GTACGAGG-3′ (Afa I recognizing site is underlined and the PAM sequence is in bold). The designed sgRNA-targeting sequences were inserted into the BbsI site of the pSpCas9(BB)-2A-GFP (PX458) vector (a kind gift from Feng Zhang, Addgene plasmid #42230) using a set of synthesized oligonucleotides as following: 5′-CACCG-CAACCCCAAGAGCTGGTACG-3′ and 5′-AAACCGTACCAGCTCTTGGGGTT GC-3′ (note that a guanine nucleotide (G, underlined), which enhances transcription of the sgRNA, was introduced at the -21 position of the sgRNA). Correctly inserted sgRNA-encoding sequences were verified by Sanger sequencing (Fasmac, Japan) using a primer 5′-ACTATCATATGCTTACCGTAAC-3′.

HEK293A cells (female origin; Thermo Fisher Scientific) were seeded in a 12-well culture plate at a density of $5 \times 10^4$ cells ml$^{-1}$ in 1 ml per well 1 day before transfection. The SMO sgRNA-encoding plasmid vector was transfected into the HEK293A cells using Lipofectamine 2000 (Thermo Fisher Scientific) according to a manufacturer's protocol. Forty eight hours later, the cells were harvested and processed for isolation of GFP$^+$ cells using a fluorescence-activated cell sorter (BD FACSDiva). The cells were sorted directly onto a cell-culture grade 96-well plate and the colonies were expanded for 20 days. Subsequently, the cells were lysed and genomic DNA was isolated with NaOH/Tris-HCl. The clones were analyzed for mutations in the targeted genes by a restriction enzyme digestion as described previously[73]. To amplify the sgRNA-targeting sites, the following pair of PCR primers was used: 5′-AAACAAGAGGCTCGTCCCTG-3′ and 5′-TAGCTGTG CATGTCCTGGTG-3′. Seven candidate clones that harbored restriction enzyme-insensitive PCR fragments were assessed for their genomic DNA alterations by direct sequencing. The two resulting, selected, sequence-determined candidate clones were further assessed for absence of SMO protein by immunoblotting (Supplementary Fig. 9). ΔSMO HEK293A cell line 3 was used in the experiments presented in this study (referred to as ΔSMO HEK293 cells).

**Cloning of receptor constructs and mutagenesis.** Nluc-A$_3$ was from Stephen Hill (University of Nottingham, Nottingham, UK). SNAP–FZD$_4$ and SNAP–FZD$_6$

were from Madelon M. Maurice (University Medical Center Utrecht, Utrecht, The Netherlands). pNluc–N1, Venus–KRas, Venus–mGsi, Venus–mGs, Venus–mG13, SMO–$Rluc8$ (coding for mouse SMO), and $FZD_4$–Nluc were from Nevin A. Lambert (Augusta University, Georgia, USA). Venus-N1 was from Addgene (#27793). In order to generate Nluc–$FZD_6$, $FZD_6$ coding sequence from SNAP–$FZD_6$ was subcloned into an empty N-terminally tagged Nluc vector containing the 5-$HT_3A$ signal peptide (from Nluc-$A_3$) using BamHI and XbaI restriction sites. First, the BamHI site present in $FZD_6$ was removed using site-directed mutagenesis (GeneArt, Thermo Fisher Scientific).Next, the $FZD_6$ sequence was cloned in-frame into the Nluc vector. FLAG–$FZD_6$–Venus and ΔCRD FLAG–$FZD_6$–Venus were subcloned from FLAG–$FZD_6$–His and ΔCRD FLAG–$FZD_6$-His, respectively, into Venus-N1 with BglII and AgeI. FLAG–$FZD_6$–Nluc and ΔCRD FLAG–$FZD_6$–Nluc were subcloned from FLAG–$FZD_6$–His and ΔCRD FLAG–$FZD_6$–His, respectively, into pNluc–N1 with BglII and AgeI. SNAP–$FZD_6$–$Rluc8$, SNAP–$FZD_7$–$Rluc8$, SMO–$Rluc8$, Nluc–DVL2, FLAG–$FZD_6$–His, ΔCRD FLAG–$FZD_6$–His, and $FZD_4$–Nluc were generated and validated in our previous studies[15,51].

The SNAP–$FZD_6$ D351A, D351E, E438A[6.54], E438D[6.54], E438N[6.54], E438Q[6.54], R442A[6.58], R442K[6.58], and K479N[7.41] mutants were made using the GeneArt site-directed mutagenesis kit (Thermo Fisher Scientific).

In order to create FRET-based sensors, the $FZD_6$ coding sequence from $FZD_6$–GFP[74] was subcloned into pcDNA3-TFP1 (Allele Biotechnology and Pharmaceuticals) between the HindIII and EcoRI restriction sites. Subsequently, the FlAsH binding sequence (FLNCCPGCCMEP) was inserted between G404 and R405 of the $FZD_6$ ICL3 using GeneArt site-directed mutagenesis kit. $FZD_6$–FlAsH–PK was generated by subcloning the $FZD_6$ insert from $FZD_6$–GFP into PK-vector (DiscoverX) with BglII and HindIII, and subsequently the FlAsH binding sequence was inserted as above. The primer sequences can be found in the Supplementary Fig. 23. All the constructs were confirmed by sequencing (GATC-Eurofins, Konstanz, Germany).

**FlAsH labeling and FRET efficiency measurements**. HEK293 cells were seeded onto coverslips. Cells were transfected 18–20 h later using Effectene (Qiagen), according to the manufacturer's instructions. Cell culture medium was replaced 24 h later and the analysis was done 48 h after transfection. For analysis of the cellular expression and FRET efficiency determination, the cells were transfected with 0.5 μg per well of the corresponding receptor construct, either $FZD_6$–FlAsH–TFP or $FZD_6$–TFP. For control experiments of basal energy transfer, cells were cotransfected with 0.3 μg per well $FZD_6$–TFP and 0.3 μg per well $FZD_6$–FlAsH–PK. FlAsH labeling was performed as previously described[75]. In brief, transfected cells were washed once with labeling buffer (10 mM HEPES, 150 mM NaCl, 25 mM KCl, 2 mM $MgCl_2$, 4 mM $CaCl_2$, 10 mM Glucose, pH = 7.3) and then incubated for 1 h at 37 °C with labeling buffer supplemented with 12.5 μM 1,2-ethanedithiol (EDT) and 1 μM FlAsH. In order to reduce nonspecific labeling, cells were rinsed once with labeling buffer and incubated at 37 °C for 10 min with labeling buffer containing 250 μM EDT. Cells were then washed twice with labeling buffer and maintained in DMEM prior to measurements.

Fluorescence imaging was performed as previously described[75]. Briefly, coverslips with FlAsH-labeled cells were mounted using an Attofluor holder and placed on a Zeiss inverted microscope (Axiovert200), equipped with an oil immersion 63× objective lens and a dual-emission photometric system (Till Photonics). Cells were maintained in imaging buffer (10 mM HEPES, 140 mM NaCl, 5.4 mM KCl, 1 mM $MgCl_2$, 2 mM $CaCl_2$, pH = 7.3), and 5 mM of BAL was added to the cells 20–30 s after the recording started. Cells were excited at 436 ± 10 nm using a frequency of 10 Hz with 40 ms illumination time out of a total of 100 ms. Emission of TFP (480 ± 20 nm) and FlAsH (535 ± 15 nm), and the FRET ratio (FlAsH/TFP) were monitored simultaneously over time. Fluorescence signals were detected by photodiodes and digitalized using an analogue-digital converter (Digidata 1440 A, Axon Instruments). FRET efficiency was calculated by inputting the maximum and minimum values of TFP into the following formula: FRET efficiency = (Δ$E$/$E$max) × 100, as previously described[75,76]. Fluorescence intensities data were acquired using Clampex software. Data were analyzed using the software GraphPad Prism 6.

**Ligand-induced changes in $FZD_6$–FRET probe**. To investigate the ligand-induced conformational changes in $FZD_6$ in populations of cells, HEK293 cells were transfected in suspension using Lipofectamine 2000 (Thermo Fisher Scientific). For the experiments $4 × 10^5$ cells ml$^{-1}$ were transfected with 1000 ng of $FZD_6$–FlAsH–TFP plasmid DNA and 100 μl of the suspension was seeded onto a poly-D-lysine (PDL)-coated black 96-well cell culture plate with solid flat bottom (Greiner Bio-One). Analysis of the cells was done 48 h after transfecting/seeding the cells using a CLARIOstar microplate reader (BMG). Following the labelling procedure described above, the cells were excited at 440–15 nm, and emission was detected at 490–20 nm and 530–20 nm. During measurements, the cells were maintained in Hanks' balanced salt solution (HBSS) containing 0.1% BSA. Recombinant WNT-5A or SAG1.3 were added to the cells 5 min after the reading started. Fluorescence changes were recorded for an additional 20 min. Data from the FRET ratio measurements obtained 2 min after the ligand addition were analyzed using GraphPad Prism 6.

**Live-cell imaging**. Confocal microscopy experiments were performed on a Leica TCS SP2 system, equipped with a HCX PL APO CS 63.0 × 1.32 oil objective. Coverslips with cells expressing the desired constructs were mounted using an Attofluor holder (Molecular Probes) and cells were maintained in the imaging buffer. TFP was excited at 458 nm and fluorescence intensities were recorded from 465–550 nm. Images were taken with 512 × 512 pixel format, 400 Hz, line average 4, frame average 3.

**NanoBRET binding assay**. ΔSMO HEK293 cells were transiently transfected in suspension using Lipofectamine 2000 (Thermo Fisher Scientific). A total of $4 × 10^5$ cells ml$^{-1}$ were transfected with a total amount of 1000 ng of plasmid DNA using Nluc–$FZD_6$: 10 ng or 100 ng for low donor condition or 1000 ng for high donor condition, and the remaining plasmid amount of pcDNA. The cells (100 μl) were seeded onto a PDL-coated black 96-well cell culture plate with solid flat bottom (Greiner Bio-One). Forty eight hours post transfection, cells were washed once with HBSS (HyClone) and maintained in the same buffer. For BODIPY–cyclopamine/10 μM SAG1.3 competition experiments at Nluc–$FZD_6$ and Nluc–$FZD_6$ R442A[6.58], and BODIPY–cyclopamine/10 μM cyclopamine competition experiments 1 ng of donor plasmid DNA was used and the experiments were performed 24 h post transfection. In the saturation experiments, the cells were incubated with different concentrations of BODIPY–cyclopamine (80 μl) for 90 min at 37 °C before the addition of the luciferase substrate coelenterazine h (5 μM final concentration, 10 μl; Biosynth #C-7004) for 6 min prior to the BRET measurement. In the competition experiments, the cells were either preincubated with different concentrations of SAG1.3 (70 μl) for 30 min at 37 °C followed by the addition of BODIPY–cyclopamine (300 nM, 10 μl); or the cells were preincubated with SAG1.3 (10 μM, 70 μl) for 30 min at 37 °C followed by the addition of the different concentrations of BODIPY–cyclopamine (10 μl). The cells were then incubated for additional 90 min at 37 °C before the addition of the luciferase substrate coelenterazine h (5 μM final concentration, 10 μl) for 6 min prior to the BRET measurement. The BRET ratio was determined as the ratio of light emitted by BODIPY–cyclopamine (energy acceptor) and light emitted by Nluc-tagged receptor (energy donor). The BRET acceptor (bandpass filter 535–30 nm) and BRET donor (bandpass filter 475–30 nm) emission signals were measured using a CLARIOstar microplate reader (BMG). ΔBRET ratio was calculated as the difference in BRET ratio of cells treated with SAG1.3 and cells treated with vehicle. BODIPY fluorescence was measured prior to reading luminescence (excitation: 477–14 nm, emission: 525–30 nm). Data were analyzed using GraphPad Prism 6.

**BRET assays**. HEK293, ΔSMO HEK293, or $ΔFZD_{1–10}$ HEK 293 cells were transiently transfected in suspension using Lipofectamine 2000 (Thermo Fisher Scientific). For the mG BRET assays, $4 × 10^5$ cells ml$^{-1}$ were transfected with 800 ng of mG plasmid DNA, 100 ng of the $Rluc8$/Nluc-tagged receptor plasmid DNA, and 100 ng of pcDNA. For the DVL2 recruitment bystander BRET assays, $4 × 10^5$ cells ml$^{-1}$ were transfected with 780 ng of Venus–KRas plasmid DNA, 200 ng of the receptor plasmid DNA, and 20 ng of Nluc–DVL2 plasmid DNA. For the direct DVL2–FZD recruitment BRET assays, $4 × 10^5$ cells ml$^{-1}$ were transfected with 800 ng of Venus-tagged $FZD_6$ plasmid DNA, 20 ng of Nluc–DVL2 plasmid DNA, and 180 ng of pcDNA plasmid DNA. The cells (100 μl) were seeded onto a PDL-coated black 96-well cell culture plate with solid flat bottom (Greiner Bio-One). Forty eight hours post transfection, cells were washed once with HBSS (Gibco or Thermo Fisher Scientific) and maintained in the same buffer. The cells were stimulated with ligands 6 min after the addition of the luciferase substrate coelenterazine h (5 μM final concentration; Biosynth #C-7004). The BRET signal was determined as the ratio of light emitted by Venus-tagged biosensors (energy acceptors) and light emitted by $Rluc8$/Nluc-tagged biosensors (energy donors). The BRET acceptor (535–30 nm) and BRET donor (475–30 nm) emission signals were measured using a CLARIOstar microplate reader (BMG). In the saturation BRET experiments, Net BRET was calculated as the difference in BRET ratio between cells expressing both donor and acceptor, and cells expressing only donor. Venus fluorescence was measured prior to reading luminescence (excitation 497–15 nm, emission 540–20 nm) and calculated as average fluorescence from each control well. Data presented in this study come from the ligand-induced BRET measurements obtained 5 min after the ligand addition (11 min after the coelenterazine h addition), and the saturation BRET measurements obtained 7 min after the coelenterazine h addition. Data were analyzed using GraphPad Prism 6.

**NanoBiT luciferase assay**. ΔSMO HEK293 cells were transiently transfected in suspension using Lipofectamine 2000 (Thermo Fisher Scientific). For the experiments $4 × 10^5$ cells ml$^{-1}$ were transfected with 100 ng of $Gα_{i1}$–LgBiT plasmid DNA, 500 ng of SmBiT–$Gβ_5$, 500 ng of $Gγ_2$, and 200 ng of receptor plasmid DNA. The cells (100 μl) were seeded onto a PDL-coated white 96-well cell culture plate with solid flat bottom (Greiner Bio-One). Forty eight hours post transfection, the cells were washed once with 0.1% BSA/HBSS (Gibco or Thermo Fisher Scientific) and maintained in the same buffer. The cells were stimulated with ligands 30 min after the addition of the luciferase substrate coelenterazine h (10 μM final concentration). Nluc$_{lum}$ (470–80 nm) was measured using a CLARIOstar microplate reader (BMG). Data from the luminescence measurements obtained 5 min after the ligand addition were analyzed using GraphPad Prism 6.

**Immunoblotting**. ΔSMO HEK293 cells were transfected in suspension using Lipofectamine 2000. For the experiments, $4 \times 10^5$ cells ml$^{-1}$ were transfected with 500 ng of receptor plasmid DNA and 500 ng of pcDNA plasmid DNA. The cells (500 μl) were seeded onto a 24-well plate. Twenty four hours later, the medium was changed and the cells were serum-starved overnight in the presence of C59 (5 nM). For the ERK1/2 phosphorylation experiments, the cells were stimulated with the indicated concentrations of SAG1.3 for 10 min at 37 °C and lysed immediately. For the DVL2 mobility shift experiments, the cells were stimulated with SAG1.3 (10 μM) for 2 h at 37 °C and lysed immediately. The lysates were obtained using urea lysis buffer (final composition: 0.5% NP-40, 2% SDS, 75 mM NaCl, 88 mM Tris/HCl, 4.5 M urea, 10% β-mercaptoethanol, 10% glycerol, pH = 7.4), initially by lysing the cells in ice-cold 1% NP-40. Lysates were sonicated and analyzed by 7.5, 10, or 4–20 % Mini-PROTEAN TGX precast polyacrylamide gels (Bio-Rad) and transferred to PVDF membranes using the Trans-Blot Turbo system (Bio-Rad). After blocking with 5% milk in TBS-T, membranes were incubated with primary antibodies in blocking buffer: rabbit anti-GAPDH (1:10000; Cell Signaling Technology #2118), rabbit anti-DVL2 (1:1000; Cell Signaling Technology #3216), rabbit anti-P-ERK1/2 (1:1000; Cell Signalling Technology #9101 S), rabbit anti-ERK1/2 (1:1000; Cell Signalling Technology #9102 S), mouse anti-SMO (1:100; Santa Cruz Biotechnology #sc-166685), and rabbit anti-SNAP tag (1:1000, New England Biolabs #P9310S) overnight at 4 °C. Proteins were detected with horseradish peroxidase-conjugated secondary antibody (1:4000, goat anti-rabbit; Thermo Fisher Scientific #31460 or 1:2000, goat anti-mouse; Thermo Fisher Scientific #31430) and Clarity Western ECL Blotting Substrate (Bio-Rad). For each experiment, the phosphorylated and shifted (PS-DVL2) to unshifted DVL2 ratios, and P-ERK1/2 to ERK1/2 ratios, were normalized by dividing each ratio by the average ratio value from all samples in each experiment as previously shown[77]. All uncropped immunoblots can be found in the Supplementary Fig. 24.

**Live-cell ELISA**. For quantification of cell surface receptor expression, HEK293 cells at the density of $4 \times 10^5$ cells ml$^{-1}$ were transfected in suspension using Lipofectamine 2000 with 1000 ng of the indicated receptor plasmid DNA or pcDNA plasmid DNA. The cells (100 μl) were seeded onto a PDL-coated transparent 96-well plate with flat bottom. Twenty four hours (Nluc-tagged constructs) or 48 h (FLAG-tagged constructs) later, the cells were washed twice with 0.5% BSA in PBS and incubated with a mouse anti-Nluc (2 μg ml$^{-1}$; RnD Systems #MAB10026) or a rabbit anti-FLAG antibody (1:1000; SigmaAldrich #F7425) in 1% BSA/PBS for 1 h at 4 °C. Following incubation, the cells were washed four times with 0.5% BSA/PBS and incubated with a horseradish peroxidase-conjugated goat anti-mouse (1:3,000; Thermo Fisher Scientific #31430) or goat anti-rabbit antibody (1:4000; Thermo Fisher Scientific #31460) in 1% BSA/PBS for 1 h at 4 °C. The cells were washed three times with 0.5% BSA/PBS, and 50 μl of the peroxidase substrate TMB (3,3',5,5'-tetramethylbenzidine; Sigma-Aldrich #T8665) were added. Following a 5 min incubation and development of a blue product; 50 μl of 2 M HCl were added and the absorbance was read at 450 nm using a Synergy 2 plate reader (BioTek). The data were analyzed in GraphPad Prism 6.

**TopFlash luciferase assay**. ΔFZD$_{1-10}$ HEK293 cells were seeded onto a 48-well plate at $2 \times 10^5$ cells per well and the next day they were transfected using Lipofectamine 2000 with M50 Super 8x TopFlash (Addgene #12456), pRL-TK Luc (Promega #E2241), SNAP–FZD$_4$, and pcDNA DNA plasmids to a total of 250 ng per well with the ratio of 2:1:1:6. 4 h post transfection, medium was changed to starvation medium together with vehicle, WNT-3A (300 ng ml$^{-1}$) or SAG1.3 (10 μM). Twenty four hours after transfection, cells were analyzed by the Dual-Luciferase Reporter Assay System (Promega #E1910) according to manufacturer's instructions in a white 96-well plate with solid flat bottom (Greiner Bio-One) with the following modifications: cells were lysed in 50 μl Passive Lysis Buffer, and 25 μl of LARII and Stop & Glo reagent were used for each well. The analysis was made on a CLARIOstar microplate reader (BMG) reading 580–80 nm for Firefly and 480–80 nm for Renilla.

**SNAP-surface Alexa Fluor 647 staining**. For quantification of cell surface expression of N-terminally SNAP-tagged receptors, ΔSMO HEK293 cells at the density of $4 \times 10^5$ cells ml$^{-1}$ were transfected in suspension using Lipofectamine 2000 with 500 ng of the indicated receptor plasmid DNA and 500 ng of the pcDNA plasmid DNA. The cells (100 μl) were seeded onto a PDL-coated black 96-well cell culture plate with solid flat bottom (Greiner Bio-One). Twenty four hours later, the cells were washed once with HBSS (HyClone) and incubated with 50 μl of 1 μM SNAP-surface Alexa Fluor 647 (New England Biolabs #S9136S) in a complete DMEM medium for 30 min at 37 °C. Subsequently, the cells were washed three times in HBSS and the fluorescence (excitation 625–30 nm, emission 680–30 nm) was read with a CLARIOstar microplate reader (BMG). Data were analyzed using GraphPad Prism 6.

**Statistical analysis**. Statistical and graphical analysis were performed using Graph Pad Prism 6 software. Data were analyzed for differences by two-tailed unpaired t-test, two-tailed one sample t-test, two-tailed paired t-test, and one-way or two-way analysis of variance (ANOVA) with Fisher's least significant difference (LSD)

post hoc analysis. Concentration–response and binding curves were fit using three, four parameters or bell-shaped nonlinear regression, and represented as a mean ± s.e.m. Specifically, the BODIPY–cyclopamine saturation binding curve was globally fit to three or four parameters nonlinear regression equations and the pK$_d$ value reported as a best-fit value ± s.d. The competition curve for SAG1.3 was globally fit using four parameters nonlinear regression and the pK$_i$ calculated with Cheng–Prusoff equation[78], and presented as a best-fit value ± s.d. Comparison between the pK$_d$ values for the NanoBRET binding curves was done using extra-sum-of-squares F test ($P < 0.05$). All the other data points throughout the manuscript represent the mean ± s.e.m. of maximum $n$ individual experiments (biological replicates) performed typically in triplicates (technical replicates) unless stated otherwise. Significance levels are given as: *$P < 0.05$; **$P < 0.01$; ***$P < 0.001$; ****$P < 0.0001$. No experimental datasets were excluded from the analysis.

**Reporting summary**. Further information on research design is available in the Nature Research Reporting Summary linked to this article.

## Data availability

Data supporting the findings of this manuscript are available from the corresponding author upon reasonable request. A reporting summary for this article is available as a Supplementary Information file. The source data underlying Figs. 2b, c, 3c, d, 4b–h, 5b, c, 6b–d, f–i and Supplementary Figs. 9d, 10a, c–f, 11b, c, 12, 13a, 14, 15a-c, 16a–e, 18a, b, 20c–e, 21a–m are provided as a Source Data file.

## Code availability

The MD trajectories used in all analyses are deposited to the GPCRmd (https://www.gpcrmd.org) with the following IDs: 196, 198, 200, and 202–205. Snapshots of the MD simulations are provided as supplementary data files.

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

## Acknowledgements

We thank Anna Krook for access to the CLARIOstar plate reader and Benoit Vanhollebeke for the $\Delta FZD_{1-10}$ HEK293 cells. Joanna J. Sajkowska-Kozielewicz is acknowledged for help in preparing the cartoon schemes. The work was supported by grants from Karolinska Institutet, the Swedish Research Council (2017-04676), the Swedish Cancer Society (CAN2017/561), the Novo Nordisk Foundation (NNF17OC0026940), Stiftelsen Olle Engkvist Byggmästare (2016/193), Wenner-Gren Foundations (UPD2018-0064), Emil and Wera Cornells Stiftelse, and the Marie Curie ITN WntsApp (grant no. 608180; http://www.wntsapp.eu). Computational resources were provided by the Swedish National Infrastructure for Computing (SNIC)–National Supercomputer Centre (NSC) in Linköping and High Performance Computing Centre North (HPC2N) in Umeå.

## Author contributions

P.K. and G.S.—conceived and designed the study. A.T. and P.K.—performed the in silico work. P.K., C.-F.B. and M.C.A.C.—performed the wet lab experiments. A.I., Y.O., P.K., M.C.A.C., J.V., J.P. and C.H.—contributed important tools and control experiments. P.K., A.T. and G.S.—prepared the figures with the input from C.H., and M.C.A.C. P.K., A.T. and G.S.—wrote the manuscript. G.S.—supervised, financed, and coordinated the project.

## Competing interests

The authors declare no competing interests.
