## [Peer Review File · Nature Communications]

Reviewers' Comments:

Reviewer #2:

Remarks to the Author:

The authors provide data that suffice for acceptance for publication in Nat Communications.

Reviewer #3:

Remarks to the Author:

This is the second revision of the manuscript by Koziellewicz et al. previously called "Structural insight into small molecule action and selectivity for Frizzled 6". Upon the manuscript transfer, the authors made additional changes (to the already extensive revision), and toned down the selectivity claims, which is a welcome development. The key advance of the study is in the demonstration of the fact that FZD6 is targetable by a small molecule, and while SAG1.3 itself is not selective, it may potentially serve as a starting point for the development of more specific probes.

Regarding selectivity: the authors demonstrated that the repertoire of receptors targeted by SAG1.3 is actually quite broad and includes, in addition to SMO and FZD6, a third receptor, FZD7. My initial suggestion of testing SAG1.3 against FZD7 was based on the observation that some of the key pocket residues in FZD7 are non-conservatively substituted as compared to FZD6, and the hope that this would be enough to abrogate SAG1.3 activity. Instead, the authors argue that the binding pocket of FZD7 is actually not that different from FZD6, and that the key residue contacts with the ligand are conserved. Upon examination of their alignment and their model, I am inclined to agree and admit that yes, this level of pocket residue conservation may result in SAG1.3 being active against FZD7 as well – which is exactly what the authors discovered, experimentally.

Regarding conclusive mutagenesis-based proof of SAG1.3 binding at the predicted pocket, it is now provided in the form of competition binding experiments using the newly developed nanoBRET assay with BODIPY-Cyclopamine.

Altogether, I believe the provided evidence is compelling and makes the study a very strong candidate for publication in Nature Communications. In addition, I encourage the authors to share models and MD trajectories with the community via GPCRMD (<https://welcome.gpcrmd.org/>), a centralized special-purpose MD deposition platform.

Rebuttal letter NCOMMS-19-905424-T

REVIEWERS' COMMENTS:

Reviewer #2 (Remarks to the Author):

The authors provide data that suffice for acceptance for publication in Nat Communications.

We thank the reviewer for the positive comments.

Reviewer #3 (Remarks to the Author):

This is the second revision of the manuscript by Koziellewicz et al. previously called "Structural insight into small molecule action and selectivity for Frizzled 6". Upon the manuscript transfer, the authors made additional changes (to the already extensive revision), and toned down the selectivity claims, which is a welcome development. The key advance of the study is in the demonstration of the fact that FZD6 is targetable by a small molecule, and while SAG1.3 itself is not selective, it may potentially serve as a starting point for the development of more specific probes.

Regarding selectivity: the authors demonstrated that the repertoire of receptors targeted by SAG1.3 is actually quite broad and includes, in addition to SMO and FZD6, a third receptor, FZD7. My initial suggestion of testing SAG1.3 against FZD7 was based on the observation that some of the key pocket residues in FZD7 are non-conservatively substituted as compared to FZD6, and the hope that this would be enough to abrogate SAG1.3 activity. Instead, the authors argue that the binding pocket of FZD7 is actually not that different from FZD6, and that the key residue contacts with the ligand are conserved. Upon examination of their alignment and their model, I am inclined to agree and admit that yes, this level of pocket residue conservation may result in SAG1.3 being active against FZD7 as well – which is exactly what the authors discovered, experimentally.

Regarding conclusive mutagenesis-based proof of SAG1.3 binding at the predicted pocket, it is now provided in the form of competition binding experiments using the newly developed nanoBRET assay with BODIPY-Cyclopamine.

Altogether, I believe the provided evidence is compelling and makes the study a very strong candidate for publication in Nature Communications. In addition, I encourage the authors to share models and MD trajectories with the community via GPCRMD (<https://welcome.gpcrmd.org/>), a centralized special-purpose MD deposition platform.

We thank the reviewer for the positive comments. We are in the process of uploading the MD files in a suitable format to the GPCRMD database.